



# High Temporal and Spatial Nitrate Variability on an Alaskan Hillslope Dominated by Alder Shrubs

Rachael E. McCaully[1,2], Carli A. Arendt[1,2], Brent D. Newman[1], Verity G. Salmon[3], Jeffrey M. Heikoop[1], Cathy J. Wilson[1], Sanna Sevanto[1], Nathan A. Wales[1], George B. Perkins[1], Oana C. Marina[1] and Stan D. Wullschleger[3]

[1]Earth and Environmental Sciences Division, Los Alamos National Laboratory, Los Alamos, 87545, United States
[2]Department of Marine Earth and Atmospheric Sciences, North Carolina State University, Raleigh, 27695, United States
[3]Environmental Sciences Division and Climate Change Science Institute, Oak Ridge National Laboratory, Oak Ridge, 37830, United States

*Correspondence to*: Carli A. Arendt (carendt@ncsu.edu)

**Abstract.** In Arctic ecosystems, increasing temperatures are driving the expansion of nitrogen (N) fixing shrubs across tundra landscapes. The implications of this expansion to the biogeochemistry of Arctic ecosystems is of critical importance, yet many details about the form, location, and availability of N from these shrubs remain unknown. To address this knowledge gap, the spatiotemporal variability of nitrate ($NO_3^-$) and its environmental and edaphic controls were investigated at an alder (*Alnus*

*viridis* spp. *fruticosa*) dominated permafrost tundra landscape in the Seward Peninsula, Alaska, USA. Soil pore water was collected from locations *within* alder shrubland growing along a well-drained hillslope and compared to soil pore water collected from locations *outside (upslope, downslope, and between)* the alder shrubland. $\delta^{15}N$ and $\delta^{18}O$ of soil pore water were consistent with the predicted range of $NO_3^-$ produced through microbial degradation of N-rich alder shrub organic matter. Soil pore water collected within alder shrubland had an average $NO_3^-$ concentration of ($4.27 \pm 8.02$ mg L$^{-1}$) and differed

significantly from locations outside alder shrubland ($0.23 \pm 0.83$ mg L$^{-1}$; $p < 0.05$). Temporal variation in $NO_3^-$ within and downslope of alder shrubland corresponded to precipitation events, where $NO_3^-$ accumulated in the soil was flushed downslope during rainfall. Enrichment of both $\delta^{15}N$ and $\delta^{18}O$ isotopes at wetter downslope locations indicate that denitrification buffered the mobility and spatial extent of $NO_3^-$. These findings have important implications for nutrient production and mobility in N-limited permafrost systems that are experiencing shrub expansion in response to a warming Arctic.

## 1 Introduction

### 1.1 Background

Ecosystems in the Arctic are directly and continually impacted by the effects of increasing global temperatures (Martin et al., 2008, Chapin et al., 2000, Hovelsrud et al., 2011), yet many details remain unknown about how vegetation and nutrient availability in permafrost systems will shift in response to climate change (Sharkhuu and Sharkhuu, 2012; Keuper et al., 2017;

Barnes et al., 2014). Because of rising temperatures, the Arctic is experiencing permafrost degradation with subsequent





feedbacks that impact soil moisture, vegetation, and nutrient availability (Shaver and Chapin, 1980; Weintraub and Schimel, 2005; Hinzman et al., 2013; Walvoord and Kurylyk, 2016). Hydrologic conditions and nutrient transport in the Arctic are controlled by the seasonal thaw of shallow active layer soils situated above the permafrost (Boike et al., 2018; Yano et al., 2010). Permafrost degradation alters landscape hydrology (Romanovsky and Osterkamp, 2000) and impacts nitrogen (N) and
carbon (C) storage and release (Schuur et al., 2015). Previous and ongoing research demonstrates the linkages that exist between N and C in arctic tundra (Weintraub and Schimel, 2003; McClelland et al., 2007; Koch et al., 2013; Street, et al., 2015; Salmon, et al., 2016; Ramm et al., 2020) and the importance of understanding both cycles.

Increased thickness of the active soil layer, thawing of previously frozen organic matter, altered soil moisture, and newly
created hydrologic pathways alter nutrient production and mobility. Deeper active layers can result in the growth and expansion of larger plant types (including shrubs) that require drier soils and deeper thaw to accommodate their root systems (Myers-Smith et al., 2011). The phenomenon of shrubs increasing in density and abundance in permafrost landscapes, is known as 'shrub expansion' or 'shrubification' (Tape et al., 2006; Weintraub and Schimel, 2003; Frost and Epstein, 2014; Ju and Masek, 2016; Myers-Smith et al., 2015; Sturm et al., 2001).


Certain shrub genera are more closely associated with climate-induced shrubification than others. Alnus viridis spp. fruticosa (Siberian alder), for example, is a shrub species (Myers-Smith et al., 2011) that fixes atmospheric nitrogen ($N_2$) through the symbiotic relationship with microbes residing within nodules in the alder root systems (Roy et al., 2007). Alders frequently establish on steep hillslopes (Myers-Smith et al., 2011; Tape et al., 2012) and have been associated with elevated concentrations
of nitrate ($NO_3^-$) in soil pore water in permafrost at high elevations like the Alps (Bühlmann et al., 2014; Hiltbrunner et al., 2014). The mobility potential of nutrients across the permafrost landscape is largely dependent on topographic gradient, which also influences vegetation, drainage, and soil redox environments (Ogawa et al., 2006; O'Donnell and Jones, 2006). Inputs of $NO_3^-$ from upland areas into surface waters can impact downslope hydrochemistry and alter downstream ecosystems (Vitousek et al., 1997; Hiltbrunner et al., 2014). Previous studies have determined that more research is needed to constrain N-fixing
vegetation and topographic controls on $NO_3^-$ mobility in the Arctic (Whigham et al., 2017; Harms et al., 2019; Harms and Ludwig, 2016).

The effects of alder on soil chemistry and stream water chemistry have been investigated in alpine and upland systems (Hurd and Raynal, 2004; Bühlmann et al., 2014; Mitchell and Ruess, 2009; Shaftel et al., 2012; Whigham et al., 2017); however,
very few of these studies have been conducted in Arctic permafrost landscapes. Rhoades et al., (2001) found that alders in permafrost soils of northwest Alaska influenced soil $NO_3^-$ and measured high foliar N in various plant types growing within alder understory, suggesting increased soil N below the shrubs. While the linkage between alders and soil N is clear, the spatial and temporal variability of soil pore water $NO_3^-$ in relation to alders, topography, redox conditions, and permafrost in the Arctic has not previously been studied in detail.



## 1.2 Study objectives

To study the influence of alder on soil chemistry in Arctic permafrost landscapes, we investigated the relationships that exist between N-fixing alders, topography, and soil moisture to determine the dominant controls on $NO_3^-$ production and availability that occur in permafrost soils on a hillslope landscape. This study was conducted as part of Next Generation Ecosystem Experiments - Arctic (NGEE - Arctic), a U.S. Department of Energy (DOE) project that informs Earth System Models (ESM) through the collection and incorporation of experimental data in the face of increasing Arctic temperatures. We focused our research on a location where a recent study estimated that alder area coverage in the hillslope shrubland community increased by 40% from 7.4 ha in 1956 to 10.4 ha in 2014, with an average rate of alder expansion of 513 $m^2$ $yr^{-1}$ (Salmon et al., 2019a). Within this field setting, we hypothesize that: 1) $NO_3^-$ availability varies both spatially (with proximity to alders) and temporally (daily and seasonally) across permafrost landscapes; 2) alder shrublands are the dominant source of $NO_3^-$ to soil pore water through mineralization and nitrification of organic matter derived from alder leaf litter and woody material; and 3) $NO_3^-$ mobility is buffered by changes in redox conditions across the hillslope topographic gradient.

## 2 Materials and methods

### 2.1 Study location

Approximately 80 km north of the Alaskan coastal town of Nome and 12 km east of Mary's Igloo, the research hillslope of interest is located at mile marker 64 of Kougarok Road (65.160714° N, -164.828275° W) and is referred to as the Kougarok Hillslope (KG Hillslope; Fig. 1a-b). KG Hillslope is a prominent hillslope with the Kuzitrin River to the north and west (~ 11° topographic slope) and a series of shallow thermokarst lakes to the east (~ 5° topographic slope). The site is composed of metagranitic Late Proterozoic bedrock outcropping at the top of the hill (Hopkins et al., 1955), and gives way to continuous permafrost on the lower slope which is likely Quaternary aged sediments composed of peat, alluvial sediment, and interlaced gravel lenses (Hopkins et al., 1955; Till et al., 2011). The hillslope is asymmetrical and slopes more steeply to the west than to the east (~ 11° and ~ 5° topographic slope, respectively). The KG Hillslope is overlain by an active soil layer containing organic peat and mineral horizons (taxonomic soil classification not available) and is well drained in the upland area due to topographic gradient. Vegetation found in the upland area primarily includes alder growing in a densely populated alder shrubland community near the crest of the hill interspersed by lichen, moss, and dwarf shrubs. Vegetation in the lowland area is characterized by either alder savanna or tussock tundra plant communities (Iversen et al., 2016; Salmon et al., 2019b).

It is important to note that the alder shrubland communities are exclusively alder, which thrive on steep hillslopes (Tape et al., 2006, 2012; Salmon et al., 2019a-b). Alder savannas occur in weakly developed water tracks and consist of short alder dispersed with other deciduous shrub species and graminoids. Between the lowland alder savanna water tracks, tussock tundra plant communities lack alder and are characterized by graminoids, dwarf shrubs, and moss (refer to Salmon et al., 2019a-b for a complete description of plant communities at this site). Active layer soils are deeper in the lowland area than the upland area



soils, have a thicker organic peat horizon, and are poorly drained due to low topographic gradient (Salmon et al., 2019a-b). The lowland is frequently saturated in inter-tussock areas, especially at the slope break between the steep upland area and shallow lowland area, and within the alder savanna community.


For our study, soil pore water compositions within alder shrubland in the upland area were compared to soil pore water compositions outside alder shrubland (both in upland and lowland areas; see Fig 1). These samples were collected temporally over four campaigns, and spatially from five transected alder patches and two lowland transects, with a higher-resolution focus and additional monitoring at two of the alder patch transects (A1 and A4) during the latter campaigns (Fig. 1; Fig. 2).

## 2.2 Sample design


To investigate the aforementioned hypotheses, this study was subdivided into two phases: an initial phase to identify $NO_3^-$ 'hotspots' in relation to the alder shrubland community (Phase 1) and a comprehensive informed phase (Phase 2) to further address each of the three hypotheses. Phase 1 (July 18-21, 2017) consisted of a synoptic survey to establish soil $NO_3^-$ variability within and adjacent to five upland alder shrubland areas (A1 – A5). To capture this variability, transects were installed with

sampling points located within alder shrubland as well as upslope and downslope of the shrubland. These transects also captured the transition between upland and lowland landscape position, which was determined to be located at the most downslope extent of each alder shrubland area.

Two additional transects were established along the eastern slope of the lowland area to capture nutrient availability in both

the alder savanna and tussock tundra communities: 1) a Road Transect that was parallel to and ~5 m from the road, and 2) a Middle Transect that was 400-m upslope of and parallel to the road transect (locations shown in Fig. 1b). Phase 2 (September 14-16, 2017, July 22-27, 2018 and September 21-22, 2018) sampling addressed the spatiotemporal variability of $NO_3^-$ within and downslope of two of the alder shrubland areas (A1 and A4 transects) that were exposed to the same topographic and climatic conditions and examined this variability with respect to topographic gradient (Fig. 1c). These two transects are located

at similar relief on the eastern slope near the crest of the KG Hillslope; however, the transects extend through and downslope of two separate alder patches allowing the nutrient dynamics associated with each shrubland area to be examined independently. These transects were examined with increasing spatial and temporal resolution throughout sampling campaigns to determine $NO_3^-$ variability within the shrubland and to capture the extent of $NO_3^-$ availability down gradient of the shrubland. To address these spatiotemporal controls, we measured soil pore water $NO_3^-$ concentrations within, between, and downslope

of alder shrubland, and measured natural abundance of $\delta^{15}N$-$NO_3^-$ and $\delta^{18}O$-$NO_3^-$ to determine sources of $NO_3^-$ and biogeochemical processes such as denitrification occurring in soil pore water across the tundra. Although an important intermediate form of nitrogen, $NH_4^+$ is not discussed with detail in this study due to low measured dissolved concentrations in these pore water samples (Table S1; Supplementary Material).



Soil pore water and bulk soil were collected from each sampling location to assess $NO_3^-$ concentrations and soil moisture. The number of samples per location for each sampling period are shown in Supplementary Material. In Phase 1 (the July 2017 campaign), samples were collected at each of the five alder patches (Fig. 1b). In Phase 2, which spanned the three subsequent sampling campaigns, all samples were collected from the A1 and A4 transects (Fig. 1c), with the goal of examining spatial and temporal variations through a tighter lens. The alder shrubland along the A1 transect covers roughly ~3,400 $m^2$ in area and

is located 20 m north of the A4 transect, which has shrubland covering roughly ~6,400 $m^2$ in area (Fig. 1c). Sample collection evolved from three sampling locations per transect in Phase 1 to sampling locations placed every 10 m along each transect in Phase 2, initiating within the alder shrubland and terminating 50 m downslope from the bottom of each shrubland area (Fig. 1c).

**2.3 Soil pore water**

Soil pore water samples were collected by installing a nest of macro-rhizons at each sample location (Rhizosphere Research Products; hereby referred to as rhizons) using the methods described by Seeberg-Elverfeldt et al., (2005) (See Supplementary Material for additional details). The soil pore water collected from each nest was integrated for a representative sample, filtered, frozen, and transported to Los Alamos National Laboratory's Geochemistry and Geomaterials Research Laboratory (GGRL) where they were stored frozen until analysis to measure major ion concentrations and isotopic compositions. Cations were

measured using inductively coupled plasma optical emission spectrometry (ICP-OES) on a Perkin Elmer Optima 2100DV instrument (Perkin Elmer Inc., USA) using United States Environmental Protection Agency (EPA) method 200.7; precision is justified to 0.01 mg $L^{-1}$. Anions were measured with ion chromatography on a Dionex ICS-2100 instrument (Thermo Fisher Scientific Inc., USA) utilizing EPA method 300 (Throckmorton et al., 2015); precision is justified to 0.01 mg $L^{-1}$. Isotopic values are reported in delta (δ) notation as the deviation from an established standard in units per mil (‰). Isotopic data for

$\delta^{15}N$ and $\delta^{18}O$ of soil pore water and soil pore water $NO_3^-$ were measured using the methods outlined in Heikoop et al., (2015); precision is justified to 0.1 ‰. A modified denitrifier method outlined by Sigman et al., (2001) and Casciotti et al., (2002) was used and analyses were made using a GV Isoprime isotope ratio mass spectrometer (IRMS) coupled to a TraceGas peripheral instrument (See Supplementary Material for additional details).

The production of $NO_3^-$ through nitrification is a microbially mediated process that produces predictable isotopic compositions through kinetic fractionation (Kendall and McDonnell, 1998). Based on the assumption that microbial nitrification utilizes two oxygen (O) atoms from water ($H_2O$) and one O atom from the atmosphere ($O_2$), the expected range of $\delta^{18}O - NO_3^-$ derived from microbial nitrification (Kendall and McDonnell, 1998) was calculated using Eq. (1):

$$\delta^{18}O - NO_3^- = \frac{2}{3}(\delta^{18}O - H_2O) + \frac{1}{3}(\delta^{18}O - O_2), \qquad (1)$$

where atmospheric $\delta^{18}O\text{-}O_2$ is +23.5 ‰ (Kendall and McDonnell, 1998). The concentration of dissolved organic nitrogen (DON) was calculated as the difference between total dissolved nitrogen (TDN-N) (measured using persulfate oxidation) and





NO$_3^-$-N (Eq. (2), assuming negligible concentrations of other inorganic N species), and the range of $\delta^{15}$N-DON was derived using Eq. (3).

$$[DON] = [TDN] - [NO_3^-],$$ (2)

$$\delta^{15}N_{DON} = \frac{(\delta^{15}N_{TDN} \times [TDN]) - (\delta^{15}N_{NO3} \times [NO_3^-])}{[DON]},$$ (3)

### 2.4 Soil and leaf litter

At each sampling location, soil was collected for soil moisture analysis from 15-cm depths (and at 30-cm depths where soil was deep enough) in pre-weighed tins, sealed with parafilm, and frozen for preservation. A total of 27 soil samples were
collected during July 2017, 6 during September 2017, and 24 in July 2018.

In July 2018, three soil pits (P1, P2, and P3) were dug and described along the A1 transect (Fig.1c). In each pit, soil was excavated to frozen soil, verified by presence of ice and sub-freezing soil temperatures. Soil was collected at an interval of 20 cm in each pit. Pits 3 and 1 had total depths of 46 cm and 55 cm to frozen soil, respectively, and soil samples were collected
at 20 cm and 40 cm. Pit 2 had a depth of 61 cm to frozen soil and soil samples were collected at 20-, 40-, and 60-cm depths. All soil samples were frozen and transported to GGRL or North Carolina State University Department of Marine, Earth, and Atmospheric Sciences in Raleigh, North Carolina where they were stored frozen until analysis (See Supplementary Material for additional details).

Alder leaf litter was collected in September 2018 from five locations along the A1 and A4 transects, stored in sealed plastic bags frozen, and homogenized prior to analysis (n=5). Thirteen soil samples and five leaf litter samples were analyzed for total N, $\delta^{15}$N of soil organic nitrogen (SON) and C/N ratios (See Supplementary Material for additional details).

### 2.5 In situ parameters

In-situ parameters were measured for each water sample and included depth to frozen soil or bedrock, soil temperature, soil
pore water pH, dissolved oxygen, and specific conductivity. Summaries of DO, pH, and conductivity are provided in Table S2 (Supplementary Material); no significant correlations with NO$_3^-$ were observed ($r^2$ = 0.02, 0.2, 0.06, respectively). Iron (Fe) speciation parameters were collected for two days during the July 2018 field campaign. These were mixed with ferrozine (to fix Fe$^{2+}$ for later analysis) on-site and subsequently analyzed at GGRL using the Fe-ferrozine method (Stookey, 1970). The Fe speciation method determined Fe$^{2+}$ and total Fe (Fe$_{Total}$; Stookey, 1970). Fe$^{3+}$ was later calculated as the difference between
Fe$^{2+}$ and Fe$_{Total}$.



## 2.6 Statistical analyses

Nonparametric Mann-Whitney rank sum tests were performed (Helsel and Hirsch, 2002) to identify significant differences between geochemical signatures - specifically $NO_3^-$, $Fe^{2+}$, $Fe_{Total}$, Mn, and $SO_4^{2-}$ - of soil pore water within the alder shrublands and outside the alder shrublands in Phases 1 and 2 (see Supplementary Material). T-tests were performed to identify significant differences between $\delta^{18}O$ during each season. P-values less than 0.05 were considered statistically significant. Simple linear regressions were used to determine relationships between $NO_3^-$ and variables including soil moisture, depth to frozen soil or bedrock, and Fe. To directly compare the variability within each chemical compound across the KG Hillslope, the coefficient of variation (Brown, 1998) was calculated for calcium (Ca), sodium (Na), chloride (Cl$^-$), and $NO_3^-$ by dividing the standard deviation of the population by the population mean (Table S3, Supplementary Material). Ca, Na, and Cl$^-$ are all compounds that typically have low variability within the environment. Consistently high coefficient of variation of $NO_3^-$ (> 1.5) and low ratios of Ca, Cl$^-$, and Na (< 1.5) indicate additional (biological) processes acting on $NO_3^-$ production. MATLAB R2017a was used for all statistical analyses and figure generation.

## 3 Results

### 3.1 Soil depth and moisture

Soils within each patch were similar in composition with general defining characteristics of a surficial peat horizon underlain by decayed peat and a transitional horizon into mineral soil at a depth of ~12 to ~15 cm, followed by bedrock or frozen soil (personal observation, Salmon et al., 2019b). During both Phase 1 and Phase 2, the mean soil moisture content (percent dry/wet weight) was lowest in the upland area and greatest in the lowland area. The lowlands were inundated during the Phase 2 September 2017 campaign and contained standing water in various inter-tussock locations during other sampling campaigns. The mean depth to bedrock or frozen soil was greater in the lowland area than the upland area during all but one sampling campaign (July 2017). (Table S4; Supplementary Material). Soil moisture content and depth to bedrock or frozen soil were not measured during the September 2018 (September 21-22) sampling campaign due to logistical and sampling challenges.

### 3.2 Phase 1: synoptic results from five alder patches

From July 18-21, 2017 soil pore water $NO_3^-$ was significantly greater (p < 0.01) within the alder shrubland than sampling locations upslope and downslope of the shrubland along the A1, A2, A3, and A5 transects (Fig. 2a; additional details in Supplementary Material). Atmospheric conditions were mild (high of ~13° C) and a brief precipitation event occurred overnight on July 18$^{th}$ (Western Regional Climate Center, 2017). Parameters including soil depth, moisture content, pH, dissolved oxygen, and conductivity did not have apparent controls on $NO_3^-$ and are reported in Tables S2 and S4 in Supplementary Material. Soil pore water $NO_3^-$ was initially negligible within the A1 alder shrubland (Fig. 3a) but increased in concentration following the rainfall event. A similar increase in $NO_3^-$ was also observed at a seep located at the transition between the upland and lowland along the A1 transect on 19 July relative to all other sampling days. This seep is likely





representative of water flowing directly from within the A1 alder shrubland to the lowland area (Fig. 2b), perhaps through fractured bedrock. Phase 1 results show that soil pore water $NO_3^-$ was elevated both from one alder shrubland patch to another (ranging from 0.3 mg $L^-$ to 58.61 mg $L^-$) and within alder shrubland relative to other sampling locations along each transect.

The four-day sample collection along the A1 transect revealed daily variations in $NO_3^-$ (0.51 to 10.08 mg $L^{-1}$) likely associated with rainfall. Both the Road and the Middle transects located in the lowland tussock tundra and alder savanna communities had low $NO_3^-$ (< 1.0 mg $L^{-1}$).

### 3.3 Phase 2: $NO_3^-$ variability between two alder patches

#### 3.3.1 September 2017

During the 2017 September sampling campaign (September 14-18), conditions were cool (high of ~5.4° C) and wet, with daily precipitation (personal observation; Western Regional Climate Center, 2017). The A1 and A4 transects were examined in greater detail by increasing the spatial resolution of soil pore water chemistry sampling (Fig. 1c). $NO_3^-$ was elevated within the A4 shrubland and downslope of the A1 shrubland (Fig. 3b), but negligible in soil pore water collected from upland areas between the two transects. The mean $NO_3^-$ concentration within the shrublands did not vary significantly from the mean $NO_3^-$

concentration directly downslope of the alder shrubland (p > 0.05; Supplementary Material). However, the mean concentration of $NO_3^-$ both within and directly downslope of the shrublands was significantly greater (p < 0.05; Supplementary Material) than the mean $NO_3^-$ concentration of the Middle and Road transects in the lowland area.

#### 3.3.2 July 2018

During the 2018 July sampling campaign (July 22-26), $NO_3^-$ was elevated along the A1 and A4 transects, both within the

shrubland and up to 20-m downslope of the shrubland, where the transition between upland and lowland occurs (Fig. 3c). Each of the five sampling days in July were dry (no precipitation, high of ~16° C), and no notable temporal variation (< 3 mg $L^{-1}$ difference) in $NO_3^-$ was observed along the A1 and A4 transects (Fig. 4a-b). $NO_3^-$ concentrations measured from two soil pits located along the A1 transect also increased with depth (Table S5, Supplementary Material). Despite the dry conditions, we directly observed water seeping from the upslope wall into each pit, suggesting that interflow was actively occurring across

the A1 transect from the upland to lowland areas. Nitrate concentrations decreased from upland to lowland along both the A1 and A4 transects (significantly different at p < 0.001; Table S5, S6 in Supplementary Material). This pattern was mirrored by Fe, which increased in concentration from upland to lowland (p < 0.001) along each transect (Fig. 4b and Table S5, S6 in Supplementary Material). Similarly, Mn increased from the upland to lowland along the A4 transect (p < 0.05; Fig. 4b and Table S5, S6 in Supplementary Material). Spatial trends in Mn were not identified, however, along the A1 transect (Fig. 4a)

with respect to Mn reduction (p > 0.05; Table S5, S6 in Supplementary Material). No trends or significant difference (p > 0.05) in $SO_4^{2-}$ concentrations were observed along the A4 transect (Table S5, S6 in Supplementary Material).



### 3.3.3 September 2018

In September 2018 (21-22), similar patterns in $NO_3^-$ concentration occurred along both A1 and A4 transects (Supplementary Material), with the highest $NO_3^-$ at locations within and directly downslope of the shrubland. Total Fe and $SO_4^{-2}$ did not vary along the A1 transect, but $Fe_{Total}$ increased along the A4 transect and $SO_4^{-2}$ increased between 0-20 m downslope of the alder shrubland (Table S5, Supplementary Material). Manganese was negligible ($< 0.01$ mg $L^{-1}$) along both A1 and A4 transects. No precipitation events occurred during sampling in September 2018.

### 3.4 Isotopes of nitrate and water

A total of 62 soil pore water samples from locations within and downslope of alder shrubland had sufficient $NO_3^-$ concentrations ($> 0.5$ mg $L^{-1}$) to measure $\delta^{15}N$ and $\delta^{18}O$ of $NO_3^-$ across all four sampling campaigns (Fig. 5; more details in Supplementary Material). Generally, $\delta^{15}N – NO_3^-$ at lowland locations downslope of the alder shrubland were more enriched than locations within alder shrubland, and high $\delta^{18}O – NO_3^-$ values correspond to high $\delta^{15}N – NO_3^-$ values at the downslope locations (Fig. 5). Because minimal fractionation takes place during $N_2$ fixation, $NO_3^-$ produced by mineralization and nitrification of organic matter derived from fixation (alder leaf litter) should have $\delta^{15}N\text{-}NO_3^-$ in the range of -5.0 ‰ to 5.0 ‰ (Kendall and McDonnell, 1998). $\delta^{15}N$ of leaf litter N (n=10) collected from the shrubland along all five transects (along with $\delta^{15}N$ data from Salmon et al., 2019b; n=5) ranged from -2.8 ‰ to -0.9 ‰. $\delta^{15}N$ of total dissolved nitrogen (TDN) ranged from -11.3 ‰ to -0.9 ‰, and $\delta^{15}N$ of dissolved organic nitrogen (DON) ranged from -19.1 ‰ to -8.0 ‰ (calculated values, see Section 2.3). $\delta^{15}N$ of SON ranged from 0.5 ‰ to 3.6 ‰ (n = 13). Atmospheric $NO_3^-$ values of $\delta^{15}N$ range from ~ -15 ‰ to ~ +15 ‰ and $\delta^{18}O$ ranging from ~ +60 ‰ to ~ +94 ‰ (Granger and Wankel, 2016). A summary of $\delta^{18}O$ compositions is provided in Supplementary Material (Table S7).

## 4 Discussion

### 4.1 Sources of nitrate

Previous studies conducted in continuous, polygonal permafrost areas without alders indicate that soil moisture is the dominant control on $NO_3^-$ production (Heikoop et al., 2015). However, at the KG Hillslope where polygonal permafrost features do not exist and the landscape is controlled by slope gradient rather than microtopography, no direct relationship was observed between and $NO_3^-$ and soil moisture ($r^2 < 0.2$; Fig. S1 in Supplementary Material). Instead of being associated with soil moisture, at this site $NO_3^-$ concentrations were tightly constrained by the presence of alder shrublands. Although $NO_3^-$ was generally absent in the poorly drained (wet) lowland area, it was elevated within alder patches on the well-drained (dry) upland area. Elevated $NO_3^-$ was not observed laterally adjacent to or upslope of alder shrubland. Therefore, while soil moisture likely plays a role in determining $NO_3^-$ availability, our results indicate that the alder shrubland community was the dominant control on $NO_3^-$ production in soil pore water on the KG Hillslope. Furthermore, Darrouzet-Nardi and Weintraub (2014) found



evidence for spatial inaccessibility of labile N in Arctic ecosystems but our findings indicate the potential for increased accessibility from the mobilization of labile N in the presence of topographic relief and precipitation.


On the KG Hillslope, isotopic signatures of $NO_3^-$ from all but two collections fall within the expected range of $\delta^{18}O - NO_3^-$ of microbially derived $NO_3^-$ (Fig. 5), further indicating that the soil pore water $NO_3^-$ is a product of microbial mineralization and nitrification rather than atmospheric deposition (Schimel and Bennett, 2004). To determine the range of $\delta^{18}O-NO_3^-$ derived

from microbial nitrification, we used Eq. (1). However, other studies suggest that $\delta^{18}O-NO_3^-$ should more closely represent $\delta^{18}O-H_2O$ (Boshers et al., 2019). We determined the range of $\delta^{18}O-NO_3^-$ derived by microbial nitrification to include both possibilities, shown in Fig. 5. If the source of $NO_3^-$ was by atmospheric deposition, values of $\delta^{15}N$ between ~ -15 ‰ and ~ +15 ‰ and $\delta^{18}O$ between ~ +60 ‰ and ~ +94 ‰ would be expected (Granger and Wankel, 2016).

In September 2017 and September 2018, some samples collected within the shrubland along the A1 and A4 transects had light $\delta^{15}N-NO_3^-$ (< -8.0 ‰; Fig. S2 in Supplementary Material). These values are likely representative of nitrification of DON derived from non-alder sources (Fig. 5; more details in Supplementary Material). For example, measurements of leaf $\delta^{15}N$ from moss, lichen, and dwarf shrubs growing upslope from the alder shrublands at the KG Hillslope range from -13.78 ‰ to -8.45 ‰ (Salmon et al., 2019b). Nitrogen from these plant types may have been mobilized downslope from the alder shrubland,

resulting in the depleted $\delta^{15}N$ found within our samples.

## 4.2 Effects of precipitation on pore water nitrate

Previous studies have linked nutrient flushing with rainfall events (Bechtold et al., 2003; Baldwin and Mitchell, 2000), and several studies have proposed that soil $NO_3^-$ inputs from mineralized and nitrified leaf litter are mobilized at wet-season onset (Yamashita et al., 2010; Bernal et al., 2003). In particular, Vink et al., (2007) found that in a forested catchment, inorganic N

accumulated within soil and leaf litter during dry periods and was subsequently flushed into headwater streams during precipitation events – an initial pulse in $NO_3^-$ in soil pore water quickly declined following increased precipitation and discharge. The dynamic temporal variability of $NO_3^-$ observed along the A1 transect (within and downslope of shrubland) associated with precipitation events in July and September of 2017 provides evidence that a similar model of $NO_3^-$ accumulation and subsequent mobilization occurs on the KG Hillslope. A 'pulse-like' signal of elevated soil pore water $NO_3^-$

was observed within and downslope of shrubland along the A1 transect (Fig. 3a) that corresponded to a precipitation event which occurred overnight between July 17 – 18, 2017. This event is reflected in the $\delta^{18}O - NO_3^-$ of two samples collected within shrubland along the A1 transect during this time period, where enriched values of $\delta^{18}O$ (> 10.0 ‰; Fig. 5) may indicate a minor influence by atmospheric deposition of $NO_3^-$.






Rainfall occurred during sampling on all three days of the September 2017 campaign (9/14/17 through 9/16/17; personal observation; Western Regional Climate Center, 2017). This precipitation may be partially reflected in the isotopic composition of water samples collected in September 2017, which had an average $\delta^{18}O$ value of -15.6 ± 0.8 ‰; ~1 ‰ more enriched than water samples collected in July 2017, July 2018, and September 18 (p<0.001; Fig. 6 and Table S7 in Supplementary Material).

During July 2018 (July 22-27) and September 2018 (September 21-22) campaigns, weather conditions were much drier (no recorded precipitation events) and $NO_3^-$ concentrations showed little variation with no daily pattern, ranging from 4.09 mg L$^{-1}$ to 4.94 mg L$^{-1}$ downslope of shrubland along the A1 transect over the five days in July 2018 and from 0.02 mg L$^{-1}$ to 0.12 mg L$^{-1}$ over two days in September 2018. While the precipitation events in July 2017 and September 2017 did appear to mobilize $NO_3^-$ from the shrubland to the lowland area (down gradient) along the A1 transect, this mobility was only observed

within the first 10-30 m downslope of the shrubland (Fig. 4a), indicating the presence of additional controls acting on $NO_3^-$ transport across the landscape.

The $\delta^{18}O$ and $\delta^2H$ signature of soil pore water approximated the global meteoric water line both in the upland soils ($r^2$ = 0.92) and lowland soils ($r^2$ = 0.81) and indicate that evaporation was negligible during sampling (Fig. S3 in Supplementary Material).

Soil pore water in the upland area was heavier in $\delta^{18}O$ composition (-15.62 ± 0.86 ‰) than in the lowland area (-16.56 ± 0.48 ‰), though these ranges overlap (0.40 ‰; Fig. S3 in Supplementary Material). The steep slopes in the upland area also promote prompt drainage of rainfall and active layer melt towards the lowlands, where the transition to a lower gradient, lack of evaporation, and limited vertical drainage result in increased residence times and increased storage capacity. Therefore, soil pore water from the lowland active layer may have had a greater component of isotopically lighter and older spring

rainfall/snowmelt and active layer ice melt, while the upland area reflected isotopically enriched, younger summer rainfall relative to our sampling campaigns (Kendall and McDonnell, 1998). Active layer ice in the less well-drained lowland areas would therefore also be expected to be lighter and to contribute more to the isotopic mass balance than in the upland area. Thus, the isotopic composition of water in upland soils likely reflects a larger summer rainfall component while lowland soils likely reflect isotopically lighter active layer ice melt (Throckmorton et al., 2015; Craig, 1961).

**4.3 Effects of redox on pore water nitrate**

The decrease in soil pore water $NO_3^-$ along the A1 and A4 transects and increase in soil moisture, $Fe^{2+}$, and Mn along the A4 transect in July 2018 (Fig. 4) and September 2018 (Tables S5, S6, in Supplementary Material) indicate a transition from an oxic environment towards a more sub-oxic environment as the slope transitions from upland to lowland. The lack of variation in $SO_4^{-2}$ across these transects indicates conditions that are not reducing enough for sulfate reduction or methanogenesis

(Jakobsen and Postma, 1999). Our data indicate that the oxic environment on the well-drained upland slope supports $NO_3^-$ production while sub-oxic environment of the poorly drained lowland area supports the reduction or denitrification of $NO_3^-$.



In a reducing environment, enrichment of $\delta^{15}N$ and $\delta^{18}O$ occurs during the process of microbial denitrification (Böttcher et al., 1990). In freshwater systems, this enrichment is typically expressed as a 1:1 ($\delta^{18}O/\ \delta^{15}N$) relationship, though many systems

have trajectories greater or less than 1 (Boshers et al., 2019). Deviations from a trajectory of 1 may be a result of nitrification and/or annamox (anaerobic ammonium oxidation) that produces $NO_3^-$ concurrently during denitrification in anoxic conditions (Granger and Wankel, 2016). The locations downslope of alder shrublands follow a similar trend of enrichment in $\delta^{15}N$ and $\delta^{18}O$ with an apparent regression slope of 0.84 (Fig. 5), suggesting denitrification occurring in the lowland areas.

Callahan et al., (2017) described upland hillslope alders as 'hotspots' for $NO_3^-$ inputs into streams at the hillslope scale, and Harms and Jones (2012) observed greater $NO_3^-$ export in soils with increased active layer thickness. These predictions are consistent with the elevated $NO_3^-$ concentrations within and directly downslope of alder shrubland along each transect (Fig. 4). The prevailing sub-oxic conditions downslope of A1 and A4 and overall scarcity of inorganic N likely restrict $NO_3^-$ production, indicating that elevated $NO_3^-$ concentrations observed at these locations was produced within the alder shrubland

patches and flushed downslope. However, mobility of $NO_3^-$ beyond the first 20-30 m downslope of shrubland along the A1 and A4 transects was not observed (Fig. 4). Inputs and subsequent dilution from flow along the gradient may be responsible for the decrease in $NO_3^-$, but it is more likely due to denitrification occurring in the sub-oxic conditions, as evidenced by the isotopic denitrification trend. Consistent with these results, a study by Harms and Ludwig (2016) predicted that saturated soils and reducing conditions may buffer N export to downslope ecosystems. Thus, while hillslopes dominated by alder will likely

increase $NO_3^-$ availability with shrubification, sub-oxic to reducing zones and soil saturation occurring at downslope locations may serve as buffers to down gradient mobility of $NO_3^-$ through microbial denitrification. Defining the role of topographically controlled redox environments on nutrient cycling in permafrost environments will be beneficial for understanding the likelihood of $NO_3^-$ mobilization to streams and subsequent transport down-gradient within N-limited landscapes.

### 4.4 Spatial and temporal variations in nitrate

Over the limited time series we collected in the 2017 and 2018 growing seasons, significant (p < 0.05; Supplementary Material) spatiotemporal variability of soil pore water $NO_3^-$ concentrations were observed at a continuous permafrost site with a distinct ridge-toe slope topography. Spatial variability existed within individual shrubland areas as well as from one shrubland area to another (Supplementary Material). These spatial differences encompassed both changes in the distribution of $NO_3^-$ within alder shrubland and changes in the magnitude of $NO_3^-$ present in the landscape over time. Temporally, we observed dynamic daily

variability in $NO_3^-$ correlated with precipitation events at locations within and downslope of an alder shrubland patch.

Although soil pore water $NO_3^-$ was elevated during dry conditions and active drainage through-flow was observed in open soil pits, the notable day-to-day changes in $NO_3^-$ during wet conditions indicate that this variability was driven primarily by the presence of rainfall. Evidence for this mechanism was observed during or following precipitation events; first after the isolated

precipitation event in July 2017 where $NO_3^-$ concentrations were elevated both within and downslope of alder shrubland in



one location (Fig. 3a), and again in September 2017 where $NO_3^-$ concentrations were elevated downslope but not within the same shrubland (Fig. 3b). On the KG Hillslope, $NO_3^-$ likely accumulates in the soil below alders as litter decomposes and gets mobilized down-gradient with the onset of rainfall. These spatial variations associated with rainfall likely indicate a 'flushing' down gradient of previously accumulated soil $NO_3^-$ from within the alder shrubland and highlight the capacity for $NO_3^-$ to be

mobilized across landscapes.

### 4.5 Future research

Findings from this study illuminate a level of complexity in N cycling that is not well understood and provides a framework for future research in this topic. Here we provide snapshots of variability and mobility of $NO_3^-$ on the KG Hillslope over brief time series. Continuous monitoring of $NO_3^-$ throughout a growing season, consideration of alder characteristics (biomass, root

and nodule density, rates of N fixation; Salmon et al., 2019a-b) and replication of this type of study are required to further characterize spatial variation within individual alder patches. Studies to further identify processes controlling the availability, transport, and fate of N in permafrost landscapes should include widespread, comprehensive investigations of N sources, total N input, organic carbon, microbial community function, and C:N relationships, such as those being conducted by Ramm et al., (2020). Future research should also emphasize the importance of characterizing N cycling to better constrain how storage and

release of C will shift with climate warming in the Arctic. Long-term impacts of $NO_3^-$ on vegetation, permafrost degradation, and interstitial water chemistry in wet downslope permafrost landscapes also require further studies. Lastly, identification of microbial communities in the sub-oxic reducing environment downslope will lend insights towards the fate of $NO_3^-$, whether it is assimilated by plants or reduced to $N_2$ or nitrous oxide gas.

The dynamic variability of $NO_3^-$ across the KG Hillslope demonstrates the importance of improving the representation of N cycle processes in ESMs. Including N and C ecological interactions in these models will decrease the uncertainty associated with forecasting atmospheric $CO_2$ concentrations in a warming climate (Thornton et al., 2009) and bolster the predictive power of future models. Although plant functional types (PFTs) are included in ESMs currently, processes such as N-fixation remain underrepresented in these classifications (Wullschleger et al., 2014). This study has shown the importance of incorporating

such processes through representation of N-fixing shrubs in ESMs.

### 5 Conclusion

Teasing apart the dynamics between alder shrubland communities, topography, permafrost, and $NO_3^-$ availability is complex because interactions between these factors vary both across time and space. The high degree of $NO_3^-$ variability observed over short timescales (days) and distances (tens of meters) has not been documented outside this study. However, we have

determined that the fixation of N within alder nodules and associated microbial plant material degradation is likely the dominant process responsible for elevated soil pore water $NO_3^-$ in proximity to alder shrublands. Nitrate present in soil within alder shrubland is likely mobilized downslope during rainfall events, however redox environments driven by hillslope



topography are important factors in buffering the spatial extent of $NO_3^-$ mobility across permafrost landscapes. These findings have implications for the anticipated nutrient responses associated with the historic and projected expansion of shrub vegetation in the Arctic and demonstrate the importance of incorporating factors such as topography, redox conditions, and plant functional type in watershed scale studies. Along with permafrost thaw (Harms and Jones, 2012) and changes in soil moisture distribution (Heikoop et al., 2015), the expansion of N-fixing shrubs across tundra landscapes is a dominant mechanism that could greatly increase $NO_3^-$ availability in the Arctic and has implications for water quality and overall ecosystem health. This study demonstrates that to fully account for the impact of shrubification on $NO_3^-$ production and export and accurately represent these dynamics in Earth System Models, scientists should include N-fixers as an independent plant functional type, and consider topographic and hydrologic gradients and the presence of geochemical reducing zones that may affect $NO_3^-$ fate and transport.

*Data Availability.* The complete data set for this research can be accessed at https://doi.org/10.5440/1544760.

*Author Contributions.* REM, CAA, BDN, JMH, VGS, and CJW conceptualized the study. REM, CAA, BDN, JMH, VGS, SS, and NAW contributed to the study investigation. REM, CAA, BDN, VGS, and JMH performed formal analyses of the data. REM, CAA, BDN, JMH, GBP, and OCM contributed to data curation of this project. CJW and SDW were involved in funding acquisition, project administration, supervision, and resource access. JMH, GBP and OCM provided laboratory resources for this project. REM, CAA, BDN, VGS, and JMH were involved in visualization, the writing of the original draft preparation and the review and editing process. CJW, SS, NAW, and SDW contributed to writing review and editing.

*Competing Interests.* The authors declare they have no conflicts of interest.

*Acknowledgements.* This research was completed with oversight and support of the Next-Generation Ecosystems Experiments (NGEE Arctic) project. NGEE Arctic is supported by the Office of Biological and Environmental Research in the U.S. Department of Energy – Office of Science. Support for this research was also provided by the Earth and Environmental Sciences Division at Los Alamos National Laboratory (LANL), the Geological Society of America, and the Marine, Earth, and Atmospheric Sciences (MEAS) Department at North Carolina State University. We would like to thank Mary's Igloo Native Corporation for their guidance and for allowing us to conduct this research on the traditional homelands of the Iñupiat people. We would like to thank Nate Conroy, Emma Lathrop, Emily Kluk, Dea Musa, and the staff at the Geochemistry and Geomaterials Research Laboratory at LANL for their assistance in fieldwork, laboratory analysis, and data organization. We thank Bob Busey for access to continuous weather data. Thank you to the entire NGEE Arctic team for their support. Finally,



we would like to thank Ethan Hyland and Gwen Hopper for laboratory assistance in the MEAS Department at North Carolina State University.

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



## Figures

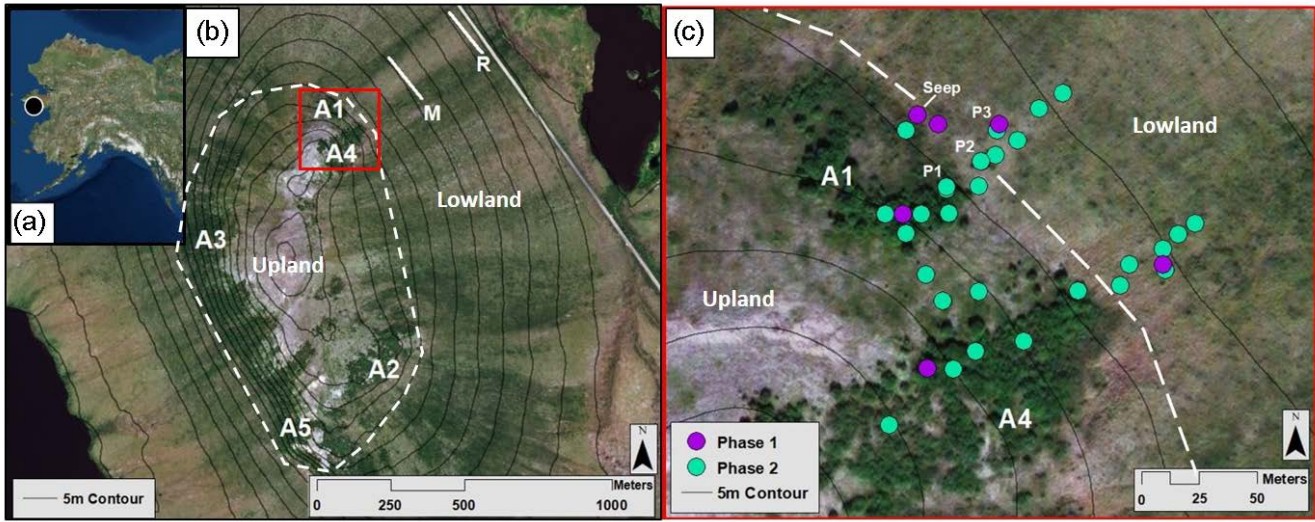

**Figure 1:** Kougarok Field Site and sampling locations. (a) Kougarok is located approximately 80 kilometers inland from the town of Nome on the Seward Peninsula, AK. (b) Alder patch and transect locations at the Kougarok Hillslope. Solid white lines represent Middle (M) and
630 Road (R) sampling transects. Dashed white line represents the boundary between upland area and lowland area. (c) Higher resolution spatial sampling locations within A1 and A4 transects; corresponds to the red box in Fig.1b. Phase 1 (July 2017) locations are denoted with purple dots. Green dots indicate additional locations sampled during Phase 2 (September 2017, July 2018, and September 2018). P1, P2, and P3 indicate Pit locations dug in July 2018. Dashed white line represents the boundary between upland area and lowland area. Aerial imagery in this figure are sourced from Esri, DigitalGlobe, GeoEye, Earthstar Geographics, CNES/Airbus DS, USDA, USGS, AeroGRID, IGN, and
635 the GIS User Community.

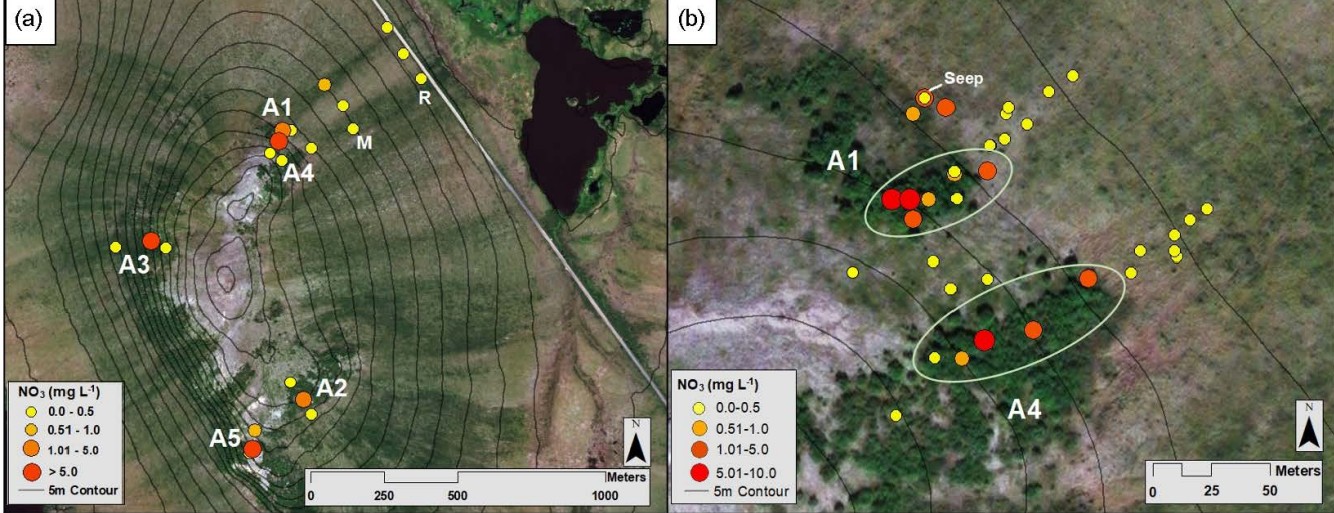

**Figure 2:** Map of mean $NO_3^-$ concentrations from the Phase 1 and 2 sampling locations where yellow indicates low concentrations and red indicates high concentrations (see Key for ranges, scales differ slightly). (a) Circles represent $NO_3^-$ concentrations at locations along the A1–A5, Middle, and Road transects in July 2017. (b) Dots represent average $NO_3^-$ concentrations along the A1 and A4 transects and between
640 the transects over all sampling campaigns. The green ellipses indicate samples collected *within* the alder shrubland, as opposed to *outside* the alder shrubland. Aerial imagery in this figure are sourced from Esri, DigitalGlobe, GeoEye, Earthstar Geographics, CNES/Airbus DS, USDA, USGS, AeroGRID, IGN, and the GIS User Community.

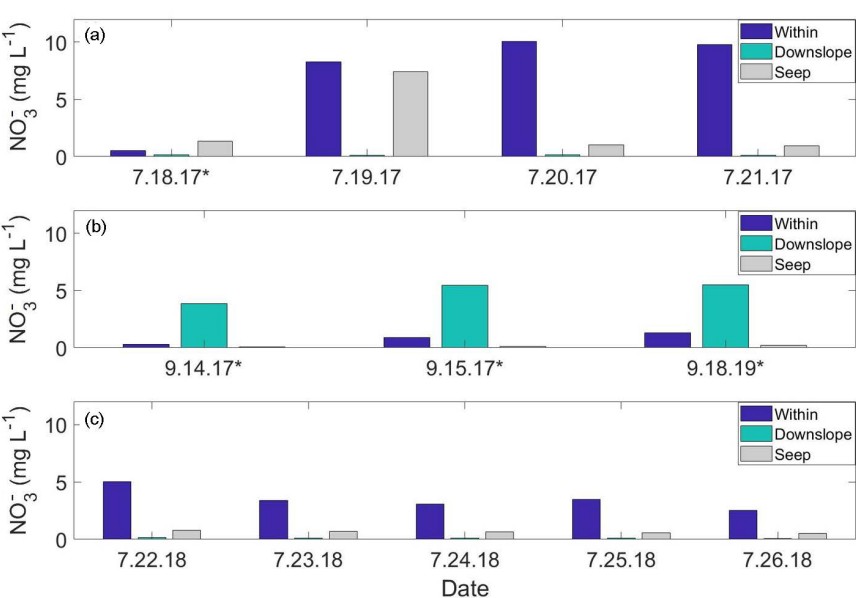

**Figure 3:** Soil pore water $NO_3^-$ time series plots from the A1 transect. 'Within' and 'Downslope' denote sample locations within and downslope of shrublands along the transect; 'Seep' denotes a seep in the ground located on the A1 transect at the transition between upland and lowland. (a) July 2017 daily $NO_3^-$ concentrations. (b) September 2017 daily $NO_3^-$ concentrations. (c) July 2018 daily $NO_3^-$. Days marked with an asterisk (*) indicate precipitation events.

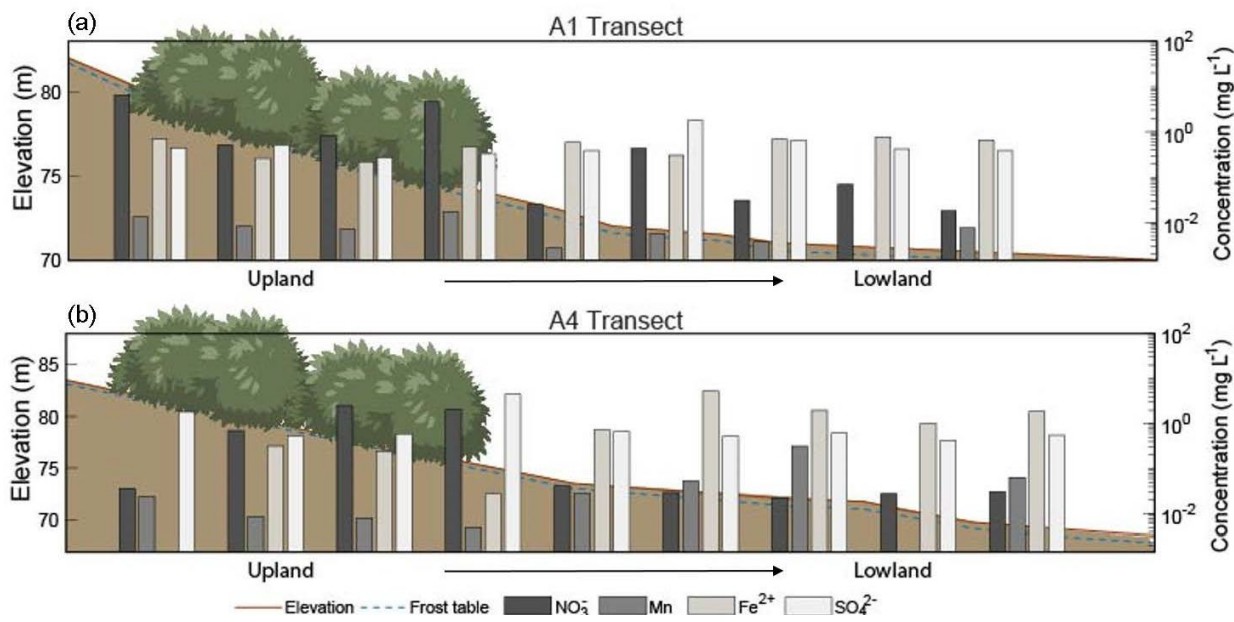

**Figure 4:** Elevation profiles and chemical concentrations along (a) A1 and (b) A4 transects, extending from the upland area to the lowland area during July (2018) sampling. Note that the elevation scale is different for A1 and A4 transects. Horizontal axis and shrubs not to scale. Depth to frozen soil or bedrock is depicted by the blue dashed line and shrubs indicate sample sites located within the alder shrubland. Redox species: nitrate ($NO_3^-$), manganese (Mn), iron ($Fe^{2+}$), and sulfate ($SO_4^{2-}$) are plotted along the secondary y-axis.




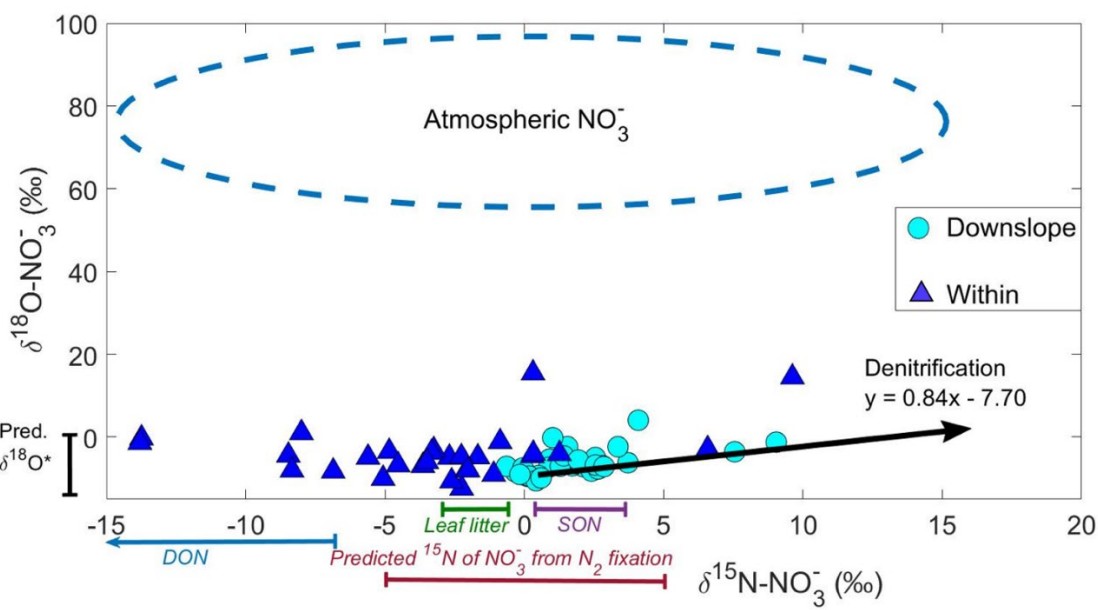

**Figure 5:** Oxygen ($\delta^{18}O$) versus nitrogen ($\delta^{15}N$) isotopes of soil pore water $NO_3^-$. Dark blue triangles represent samples within alder shrubland in upland areas. Light blue circles represent samples downslope of alder shrubland in lowland areas. The denitrification process of the downslope samples is denoted by the black arrow inside the plot (y = 0.84x - 7.70). The vertical black line segment (Pred. $\delta^{18}O^*$) denotes the predicted range of $\delta^{18}O$ of $NO_3^-$ produced by microbial nitrification (Eq. 1). The horizontal red line segment indicates the likely $\delta^{15}N$ of $NO_3^-$ range for mineralization and nitrification of organic matter derived from alder material (N-fixation; Kendall and McDonnell, 1998). The likely range of $\delta^{15}N$ of $NO_3^-$ from leaf litter is shown by the horizontal green line segment. The horizontal blue arrow denotes the range of $\delta^{15}N$ of $NO_3^-$ from DON, and the horizontal purple line segment indicates the range of $\delta^{15}N$ of $NO_3^-$ from SON.

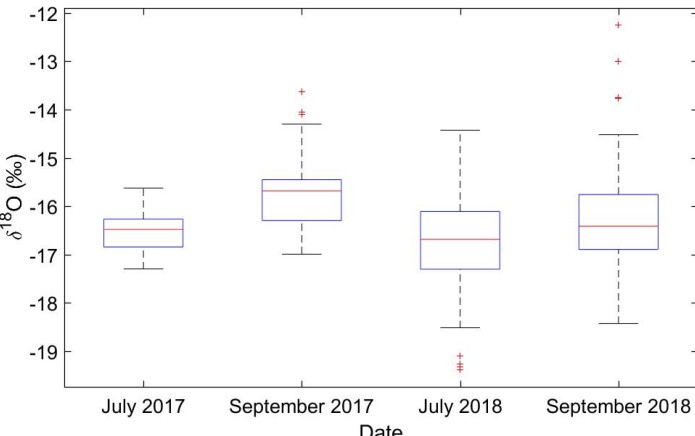

**Figure 6:** Oxygen isotopes ($\delta^{18}O$) of soil pore water on the KG Hillslope during July and September of 2017 and 2018. The red line within each box represents the sample mean and red crosses indicate outliers. *In September 2017, the mean $\delta^{18}O$ (-15.6 ± 0.8 ‰) was heavier than $\delta^{18}O$ in July 2017, July 2018, and September 2018 (p < 0.05) and likely reflects the precipitation event that occurred continuously over the September 2017 sampling campaign.