# Peer review of "High Nitrate Variability on an Alaskan Permafrost Hillslope Dominated by Alder Shrubs"

_The Cryosphere, 2021_

## Referee Comment (RC1)

**Reviewer comment to Arendt and ten co-authors on: High temporal and spatial nitrate variability on an Alaskan hillslope dominated be alder shrubs**

The topic of the study to investigate the N cycle, in particular shifts in the nitrate concentration in the soil pore water in permafrost soils, affected by the $N_2$ fixation of alder (*Alnus viridis* ssp. *fruticosa*) is generally interesting. However, the study and the structure of the article has some major flaws and is a listing of observations instead of presenting a study with a clear focus. The structure of the article may be improved, but -unfortunately- several limitations of the design of the study cannot be corrected anymore (in retrospective).

*Aspects to can be improved:*

Many aspects that emerge in the result section, are not adequately described in the M&M section. Some descriptions given in the Supplementary Material

- "Seep" has not been explained in M&M
- Patch of Alnus shrubland: dimension/ size of the patch not defined. Valuable pieces of information would be height of bushes, density of alder branches per m2. Please add.
- Authors present the total numbers of measuring points or samples (including soil pits), but it remains unclear what is really a replicate; a transect contains how many measuring points? The structure of the measuring points is also key for the statistical analysis. Both, temporal and spatial variability has to been taken into account in the statistical analysis. Here, one gains the impression, that one and then the other has lumped together. To emphasize that non parametric versus t-tests have been used is not adequate here.
- The term "Alnus savanna" simply does not exist and is misguiding, please remove this term throughout the whole text!

*Flaws in the design (cannot be correct anymore)*

- The nitrate concentrations in soil pore water of Alnus shrubland along the hill is compared to those in the soil pore water in the lowland. However, authors stated that the soil was partly covered by standing water and the alder bushes were much smaller in the lowland area or do not even exist in the lowland (Figure 4!) So, standing water on one hand may dilute the nitrate concentrations and create denitrifying conditions, and on the other hand smaller Alnus shrubs have for sure a lower $N_2$ fixation. These two aspects cannot be disentangled. So, it is obvious that the nitrate concentration is lower under such conditions, described here as major result (please avoid overstatements in general). Furthermore, it has already been shown that the nitrate concentrations in the soil water under Alnus stands is much higher than under non-Alnus stands due to the $N_2$ fixation (not a new result!)
- On site weather station is fully missing. Although authors stated precipitation events as key for leaching processes.
- A measuring campaign during 4 days (year 2017) is simply not representative for a seasonal measure (overstatement).

*Detailed comments:*

Abstract: Line 11: in the Arctic with capital letter, but arctic ecosystems with small letter. Please adapt throughout the MS.

Line 13: Simply not true, please consult the literature and adapt

Line 14: Edaphic controls for the nitrate concentrations has not been shown.

Line 18: I do not agree that all the nitrate is produced by degradation of N-rich alder shrub organic matter…or depends on your definition of organic matter..Are the corolla structure of the alder roots/nodules where the N2 fixation takes place, organic matter (I think not!)? The majority of the nitrate is already released during the N2 fixation.

Line 19, etc.: In general, better to express the nitrate concentrations as nitrate-**N** (enhanced comparability to other N compounds, atmospheric N deposition)

Line 23: denitrification buffers nitrate mobility. Strange description! Nitrate is transformed into N2 (complete) or N2O (incomplete) denitrification. From an ecological point of view the production of N2O is worse than nitrate mobility! Please adapt.

Line 24. Nutrient production is a misguiding term. Through N2 fixation nitrogen as a nutrient gets available, then it is transformed or lost again through complete denitrification..please adapt.

Line 40: Nutrient availability instead of nutrient production

Line 46: Latin names - italic

Line 48: instead of microbes, add here: *Frankia* bacteria

Line 51, line 57: Wrong!! Brühlmann et al. 2014, Hiltbrunner et al. 2014: Both studies are located in the montane (not in the alpine vegetation belt of the Alps), but clearly not on permafrost soils! Increase of alder shrubland due to changes in land use, not increasing temperatures. Please correct and add this aspect of land use changes.

Line 68-70: Necessary?

Line 73-76: Hypotheses rather weak as already widely known that alder shrubs through their N2 fixation are source for nitrate. And your measuring campaigns cover some days in July and September, not seasons. And see comments on line 18 and 23 (comments are not repeated here).

Line 80: unusual format for coordinates, add elevation of the KG hillslope, please adapt

Line 95ff, add species names of the dominant species of graminoids and dwarf shrubs.

Line 101-112: rather unclear and wordy description. Be more precise here! Avoid expression such as initial phase and comprehensive informed phase (rather empty expressions).

Line 114-135 unclear what means additional transect here, how many sampling points per transect?? Shorten! Please be more precise.

Line 140ff: a nest of macro-rhizon: please define in the main text (not in the Supp. Material)- I wonder how long lasted the installation, for such short sampling intervals (of 4 days) the installation duration may affect the water sampling of the first day. Please explain!

Line 167. Unclear description, which transect (?) and soil 0-15cm has not been sampled?

Line 180: Why five litter samples when A1 and A4 have three sampling locations each, unclear…

Line 184: instead of each water sample…*In situ* parameters were measured for each water sampling location …

The whole M & M section needs to strongly streamlined, now it is a potpourri of very different measurements and reader often does not know why for what purpose a measurement was carried out, besides when and how many times…I suggest to present all these different locations and campaigns in a Table.

Line 191-203 Statistical analysis: Weak description, no information how normal distribution, outliers, etc were handled. Weird description of processes acting on nitrate production.

Line 205: I assume that soils (and correspondingly patches) along the slope differ from soils in the inundated lowlands. And permafrost occurrence and thickness of the layer- were they similar along the hillslope as in the lowlands? Please specify!

Line 208: personal observation of whom? Mean gravimetric soil moisture content

Line 209: unclear: which other sampling campaigns?

Line 212: what are logistical and sampling challenges? Unclear.

Line 213: adapt subtitle, 4 measuring days do not allow to delineate synoptic results (overstatement)

Line 214: you mean.. nitrate concentration was higher … please adapt

Line 216: Strange description on atmospheric condition, air temperature, mean or maximum , please be precise, brief precipitation event, add where this has been measured and the exact rain amount.

Line 220: Seep? Not introduced in M& M.

Line 221: relative to the other three sampling days in July

Line 222: avoid such blue sky interpretation! Adapt

Line 224: 58.61 mg L-1 Typing error, such high values are not presented in the Figures nor in Tables!

Line 230: please describe the weather conditions during your campaigns properly

Line 234-237:  two times the same results with very different outcomes? Rather weird description, please improve.

Line 228-258: Not fully clear what you like to present as results here, rather repetitive description on the different campaigns, not really convincing, please improve

Line 259-271: Rather a potpourri of observation, in M &M section you mentioned 5 litter samples, now there are 10. What do these results tell you.

Discussion: Obvious that the N2 fixation is the main source for the nitrate! So, please reorganise the whole discussion, now it reads like another result section! Please adapt

Line 325-326: what kind of additional controls? Please avoid such empty sentences.

Line: 329: A new aspect emerges: the comparison with the global meteoric water line! Why do you expect evaporation ins such a wet landscape? See line 244 Water flows even during periods without rain…

Line 386: Future research? Rather bizzare that the authors list all the requirements for a more solid study. I would not declare these points as future research but as prerequisites for the current study!

Line 401: Really unclear how such a single hillslope study should be of value for ESM

Line 4902: bolster??

Conclusion: Already established that alder fix atmospheric N and therefore contribute to higher nitrate concentration in soil water. Rather redundant conclusions.

**Figures**

Figure1 (a) better to insert a map

Figure 2 a, b: Redundant to use different size of symbols and different colours. Please use for the same nitrate concentration the same colour! It is not really convincing to present means AND single values

Figure 3: add sd (of bars) and add weather conditions (at least air, soil temperature and rain in mm)

Figure 4: There are no alders in the lower part of both transects, correct? I have some doubts whether the log Y scale really helps here. Table with the values (mean ± sd, number of replicates would be for sure more informative)

Figure 6. Boxplots largely overlap, that means no significant differences. How did you get this $P<0.05$? Though single t-tests? (Multiple mean test would be correct). Add n here…

---

## Referee Comment (RC2)

**Review comments of the manuscript tc-2021-166 by McCaully, Arendt et al. titled: High Temporal and Spatial Nitrate Variability on an Alaskan Hillslope Dominated by Alder Shrubs.**

**General comments**

This study presents a comprehensive dataset which illustrates how substrate source (alder litter) and spatial connectivity in a sloping permafrost landscape may be larger control on $NO_3^-$ presence in uplands than soil moisture content, how increased $NO_3^-$ related to N fixer presence may be mobile in the landscape, and how redox conditions related to soil moisture and topography impacts the spatial extent of this mobility. This is a valuable contribution which underlines the importance of considering topography and N fixers as plant functional type in predictions of future plant N availability and potential $N_2O$ emissions.

However, this is a complicated dataset, and the study could benefit from a more coherent storyline, where the different datasets are presented not only in sequence, but are used together to tell a common story and the reader understands why the methods were chosen. This is done nicely in the abstract, but lacks in the discussion, where the $\delta^{18}O$ results and the $\delta^{15}N$ and $NO_3^-$ concentration results are, I suspect, not intended to be two separate stories, but they appear as such at the moment.

A few more specific section comments:

Introduction: The language could use an overhaul, mainly a condensation of the text, where some points are repeated and some sentences/sections come out of context (see specific comments below).

Materials and methods: The sample design is very comprehensive and complicated and as such benefits from a detailed description. However, the information could be more closely related to Figure 1b and 1c for clarity and condensed. The description of isotopic calculations is clear and useful. There is a lack of a quantitative estimate of precipitation (now currently addressed simply as "Precipitation events") from e.g. a micrometeorological station, as the precipitation downslope movement of $NO_3^-$ is such a central part of the results.

Discussion: The storyline of the discussion is not clear and appears more as a list of results related to literature than a use of results to illuminate your research questions. As an example, the discussion of $\delta^{18}O$ related to precipitation events (line 328-339) comes a bit disconnected from the $NO_3^-$-story, but I suspect there is a point related to N transport, which needs to be clarified. The discussion needs to be restructured and condensed to tell the study story based on the results.

Because of the large revisions needed in the communications of the results, I recommend that the manuscript can be reconsidered after major revisions.

**Specific comments with line numbering**

Line 35-37: Which links and why is it important? Give one or two examples for a more engaging story.

Line 51-56: This second half of the paragraph seems a bit out of place, because the text introduces alder effects on soil chemistry above and continues below. Consider moving it and even skipping line 51-52 or replacing the sentence in line 39-40 as they say much the same.

Line 68: Alternatively "situated in a hillslope landscape"

Line 91-91: I don't understand the function of this sentence in relation to the next sentences.

Line 160: A sentence on how $\delta^{18}O$ from $H_2O$ (soil solution) in your $NO_3^-$ is derived would be useful here.

186-190: Iron, sulfate and Manganese enter the story a bit abruptly here. If they have a function in the study design (as it is later clear that they have), please add a sentence earlier when explaining the study scope and strategy, adding the function of measuring those parameters.

Line 269: You define all the other pools, but SON is not defined (Soil Organic Nitrogen, I assume)?

Line 284: This is an interesting finding from this study

Line 290-91: This statement, referring to Boshers et al. (2019) should be explained further. While the equation 1 is nicely explained previously, the argumentation for choosing this method should be discussed in relation to alternatives. You mention that "both possibilities" (line 291-292) are shown in figure 5. By this, I take that you mean the $H_2O$-derived only and the Eq. 1 determined predictions (?), however, I see only one interval of predictions in figure 5. The text and the link needs a better explanation.

Lines 328-339: This section is interesting and coherent in its argumentation, but its place in the story of the manuscript is not clear. The point may be that there is a connection to the $NO_3^-$ transport and –source, however, this link needs to be clearer for this section to be relevant to the overall story.

Line 375-385: This section is a good example of clear, well-written communication/discussion of the results. !

**Technical comments with line numbering:**

Line 19: The parentheses around $NO_3^-$ concentrations are not necessary and should either be removed or the sentence restructured

Line 32: Consider using "near-surface hydrologic conditions" in order to exclude e.g. subpermafrost groundwater

Line 181: a comma is likely missing between "bags" and "frozen".

Line 184: Soil temperature at which depth?

Line 185: Introduce DO as Dissolved Oxygen before abbreviating

Line 200-201: Back up this statement with a reference?

Line 214: The beginning of this sentence should be reformulated – for once, the comma seems misplaced before "2017"

Figure 4: the lower part of the figure is cut off by the caption

---

## Author Comment (AC1)

**Author response to Anonymous Referee #1 comments: High temporal and spatial nitrate variability on an Alaskan hillslope dominated be alder shrubs**

*All author responses appear in grey italics below specific comments from Anonymous Referee #1.*

The topic of the study to investigate the N cycle, in particular shifts in the nitrate concentration in the soil pore water in permafrost soils, affected by the $N_2$ fixation of alder (*Alnus viridis* ssp. *fruticosa*) is generally interesting. However, the study and the structure of the article has some major flaws and is a listing of observations instead of presenting a study with a clear focus. The structure of the article may be improved, but -unfortunately- several limitations of the design of the study cannot be corrected anymore (in retrospective).

*The authors thank Anonymous Referee #1 for their feedback and agree that there are some limitations of the data presented due to the disparate nature of field access and sampling campaigns. While a continuous time-series would have been preferable, we were limited due to scheduling permissions and availability but firmly believe there is inherit value in the data that we were able to collect and the following text details our motivation for the various campaigns. Since there had been no actual measurements of nitrate concentrations in soil pore water of Arctic alder stands at the time this investigation began, we conducted an initial reconnaissance study in year 1 to gain insights to the distributions and concentrations of pore water nitrate. We augmented these data with additional targeted sampling campaigns in the following year but were limited on timeframes of the follow-up investigations.*

***Aspects to can be improved:***

Many aspects that emerge in the result section, are not adequately described in the M&M section. Some descriptions given in the Supplementary Material

- "Seep" has not been explained in M&M
  *The authors thank Anonymous Referee #1 for this suggestion. The seep was initially defined in the Figure 3 caption but we will add the following definition in the main text in the next version. "The seep sampling location is defined by a direct seep from the ground located on the A1 transect at the transition between upland and lowland zones. The volume of water coming from this seep was too small to measure directly but is estimated to be $< 2$ cm$^3$ s$^{-1}$. However, water was actively trickling from this location during all sampling campaigns and is likely representative of active layer melt that surfaces at the upland-lowland transition."*

- Patch of Alnus shrubland: dimension/ size of the patch not defined. Valuable pieces of information would be height of bushes, density of alder branches per m2. Please add.
  *The authors thank Anonymous Referee #1 for this suggestion. Figure 1c visually shows the scale of the alder stands we investigate but we will incorporate respective sizes of A1 (~2500 m$^2$) and A4 (~11250 m$^2$) into the main text for reference. Additionally, Salmon et al. (2019) identifies the alder stand height (cm), basal area per shrub (cm$^2$ stem), basal area (cm$^2$ m$^{-2}$ ground), nodule biomass (g m$^{-2}$ ground), and nodule biomass (g m$^{-2}$ stem) in their study that include the 'alder shrubland' and 'alder savanna' regions that we investigate here. If accepted, we will also add these details to our manuscript.*

  *Reference:*

  *Salmon VG, Breen AL, Kumar J, Lara MJ, Thornton PE, Wullschleger SD, Iversen CM. 2019. Alder Distribution and Expansion Across a Tundra Hillslope: Implications for Local N Cycling. Frontiers in Plant Science 10: 1–15.*

- Authors present the total numbers of measuring points or samples (including soil pits), but it remains unclear what is really a replicate; a transect contains how many measuring points? The structure of the measuring points is also key for the statistical analysis. Both, temporal and spatial variability has to been taken into account in the statistical analysis. Here, one gains the impression, that one and then the other has lumped together. To emphasize that non parametric versus t-tests have been used is not adequate here.

  *The authors thank Anonymous Referee #1 for this suggestion and believe some clarifying details will provide the needed context for interpretation of the results.*

  *At present, Figure 4 visually represents the relative locations and number of distinct rhizon clusters along the A1 and A4 transects, this information can be found in more detail in Table S5 in the Supplement, which identifies each specific sampling location that is clustered within the different sampling regions (i.e. A1_WI_Up, A1_WI_Mid, and A1_WI_Down, represent three distinct sampling locations that are located within Alder Stand 1 (A1)).*

  *Table S6 in the Supplement displays the setup and results from the Mann Whitney rank sum tests performed for each constituent. The intention of the Mann Whitney statistical analyses is to look at the difference between 'upland' and 'lowland' sampling locations (spatial differences) and not between seasons (not temporal differences). Thus, here 'n' is the total number of samples collected in each location (upland and lowland) from both July and September 2018 combined. Based on advisement from a statistician consult, this approach appeared to be the best way to directly compare differences in 'upland' and 'lowland' sampling sites overall. The individual rhizon nests that make up 'upland' and 'lowland' are defined in Table S5.*

  *After reviewing these comments and discussing with coauthors who performed the statistics, we have decided that if this manuscript is accepted for publication, we would modify Supplement Tables S5 and S6 to improve clarity and add text to the main manuscript to detail that the Mann Whitney statistical tests performed on nutrient data were not intended to highlight temporal variation, but to identify significant differences between the upland and lowland site geochemistry regardless of season.*

  *Because of the discontinuous nature of our sampling campaigns, our data is not suited for temporal comparisons of nitrate but we were interested in the influence of seasonality and precipitation events on water isotopic signatures. Thus, the only temporal statistical analysis performed was on the water isotopes (Table S7), which looked at differences between each season and the overall mean of $\delta^{18}O$ of water (not nitrate). The total number of samples collected within each campaign define 'n' for each campaign (samples not grouped spatially). These samples are not divided into 'upland' and 'lowland' samples but are lumped holistically to show seasonal trends as opposed to spatial trends. The 'n' varies widely between months because the number of days per campaign varied.*

- The term "Alnus savanna" simply does not exist and is misguiding, please remove this term throughout the whole text!

  *The authors thank Anonymous Referee #1 for this comment but respectfully disagree. The authors would like to note that we use the term 'alder savanna', not 'alnus savanna'. The term 'alder savanna' is defined by Frost et al., (2013), who write, "Such shrubland communities, colloquially referred to as 'alder savannas', have been described at several locations in Low Arctic and interior montane Alaska (Racine 1976, Racine and Anderson 1979, Chapin et al., 1989). Regular spacing of alders in 'alder savannas' has been attributed to intra-specific competition for limiting nutrients (Chapin et al., 1989)."*

*A coauthor on this paper (Salmon) also uses this 'alder savanna' for this community in Salmon et al. (2019) and Ben Sulman's recent modeling paper (Sulman et al., 2021).*

*References:*

*Chapin FS, McGraw JB, Shaver GR. 1989. Competition causes regular spacing of alder in Alaskan shrub tundra. Oecologia 79: 412–416.*

*Frost GV, Epstein HE, Walker DA, Matyshak G, Ermokhina K. 2013. Patterned-ground facilitates shrub expansion in Low Arctic tundra. Environmental Research Letters 8: 015035.*

*Racine C H 1976 Flora and vegetation Biological Survey of the Proposed Kobuk Valley National Monument. Final Report ed H R Melchior (Fairbanks, AK: Alaska Cooperative Park Studies Unit, Biology and Resource Management Program, University of Alaska) pp 39–139*

*Racine C H and Anderson J H 1979 Flora and vegetation of the Chukchi-Imuruk area Biological Survey of the Bering Land Bridge National Monument: Revised Final Report ed H R Melchior (Fairbanks, AK: Alaska Cooperative Park Studies Unit, Biology and Resources Management Program, University of Alaska) pp 38–113*

*Salmon VG, Breen AL, Kumar J, Lara MJ, Thornton PE, Wullschleger SD, Iversen CM. 2019. Alder Distribution and Expansion Across a Tundra Hillslope: Implications for Local N Cycling. Frontiers in Plant Science 10: 1–15.*

*Sulman BN, Salmon VG, Iversen CM, Breen AL, Yuan F, Thornton PE. 2021. Integrating Arctic Plant Functional Types in a Land Surface Model Using Above- and Belowground Field Observations. Journal of Advances in Modeling Earth Systems 13: e2020MS002396.*

*The authors will add in these references with additional clarifying text to avoid any misguiding caused by the use of 'alder savannas'.*

**Flaws in the design (cannot be correct anymore)**

- The nitrate concentrations in soil pore water of Alnus shrubland along the hill is compared to those in the soil pore water in the lowland. However, authors stated that the soil was partly covered by standing water and the alder bushes were much smaller in the lowland area or do not even exist in the lowland (Figure 4!) So, standing water on one hand may dilute the nitrate concentrations and create denitrifying conditions, and on the other hand smaller Alnus shrubs have for sure a lower $N_2$ fixation. These two aspects cannot be disentangled. So, it is obvious that the nitrate concentration is lower under such conditions, described here as major result (please avoid overstatements in general). Furthermore, it has already been shown that the nitrate concentrations in the soil water under Alnus stands is much higher than under non-Alnus stands due to the $N_2$ fixation (not a new result!)

  *The authors thank Anonymous Referee #1 for this comment and will make sure our field location descriptions are clarified to accurately represent the conditions present in the uplands versus the lowland sampling sites. The lowland sampling sites are comprised of alder savanna, where alder are present but not in dense stands as they are in the uplands. The findings observed provide insight to the transport and mobility potential of nitrate from dense alder stands in upland environments: topography/gradient and precipitation/moisture in tundra environment with alders largely control this potential. So although nitrate availability may increase directly in the soil-pore water of dense alder stands, the moisture/precipitation and topography/gradient conditions of the alder environment likely limit/control of the mobility potential of this critical nutrient. The authors will ensure they properly clarify this point and critically assess the major results discussion and remove instances of overstating outcomes.*

- On site weather station is fully missing. Although authors stated precipitation events as key

for leaching processes.

*A weather station has been installed for continuing studies at this NGEE field location but was unavailable at the time of the study presented here. We agree that measurements provided by a weather station would have been useful but are unable retroactively obtain this information. Weather records of towns ~50 km away from the field site are accessible and referenceable but not a direct representation of the weather conditions at our remote field site. Field observations of precipitation events are included for additional context of trends and variability observed. While we are not able to quantify the extent of precipitation events, we are able to identify trends correlating with observed precipitation events that provide insight to nutrient transportation responses to weather observed.*

- A measuring campaign during 4 days (year 2017) is simply not representative for a seasonal measure (overstatement).

  *The authors agree with Anonymous Referee #1 and will modify language accordingly to avoid temporal overstatements. While we do not have a full seasonal time series, the multiple short time-series we have captured provide valuable insights to snapshots of compositions and variations that may exist within short timeframes within specific seasons.*

**Detailed comments:**

Abstract: Line 11: in the Arctic with capital letter, but arctic ecosystems with small letter. Please adapt throughout the MS.

*The authors thank Anonymous Referee #1 for pointing this inconsistency out and the text will be updated accordingly.*

Line 13: Simply not true, please consult the literature and adapt

*While some details are known, many are not. The authors stand by this statement but will modify the text to include additional details of where knowledge gaps exist (ex: variable nature and transport potential of nitrate) and where further characterization is needed.*

Line 14: Edaphic controls for the nitrate concentrations has not been shown.

*'Edaphic' will be deleted from the text.*

Line 18: I do not agree that all the nitrate is produced by degradation of N-rich alder shrub organic matter…or depends on your definition of organic matter..Are the corolla structure of the alder roots/nodules where the N2 fixation takes place, organic matter (I think not!)? The majority of the nitrate is already released during the N2 fixation.

*The authors thank Anonymous Referee #1 for this comment and will modify the language here to include text referencing the role of root nodules in nitrate production.*

Line 19, etc.: In general, better to express the nitrate concentrations as nitrate-**N** (enhanced comparability to other N compounds, atmospheric N deposition)

*The authors will change references of nitrate to references of nitrate-N.*

Line 23: denitrification buffers nitrate mobility. Strange description! Nitrate is transformed into N2 (complete) or N2O (incomplete) denitrification. From an ecological point of view the production of N2O is worse than nitrate mobility! Please adapt.

*The authors are not claiming that the production of N2O isn't ecologically important, we are stating that nitrate-N is unable to be transported if it is converted to other N-species. We will modify the text to read 'denitrification limits the mobility of nitrate-N by transforming it to other N-species.'*

Line 24. Nutrient production is a misguiding term. Through N2 fixation nitrogen as a nutrient gets available, then it is transformed or lost again through complete denitrification..please adapt.

*The authors will alter text from 'nutrient production' to 'nutrient availability' here and in the remaining text.*

Line 40: Nutrient availability instead of nutrient production
*The authors will alter text from 'nutrient production' to 'nutrient availability' as suggested.*

Line 46: Latin names – italic
*The authors will verify that all Latin names are in italic before any resubmission occurs.*

Line 48: instead of microbes, add here: *Frankia* bacteria
*The authors thank Anonymous Referee #1 for this suggestion and will alter the text from 'microbes to 'Frankia bacteria' as suggested.*

Line 51, line 57: Wrong!! Brühlmann et al. 2014, Hiltbrunner et al. 2014: Both studies are located in the montane (not in the alpine vegetation belt of the Alps), but clearly not on permafrost soils! Increase of alder shrubland due to changes in land use, not increasing temperatures. Please correct and add this aspect of land use changes.
*The authors thank Anonymous Referee #1 for this observation and will modify this section accordingly to appropriately cite these references and identify land use change as an additional driver of shrub expansion.*

Line 68-70: Necessary?
*The authors will delete this text from the Study Objectives section because these details are already included in the Acknowledgements section.*

Line 73-76: Hypotheses rather weak as already widely known that alder shrubs through their N2 fixation are source for nitrate. And your measuring campaigns cover some days in July and September, not seasons. And see comments on line 18 and 23 (comments are not repeated here).
*The authors thank Anonymous Referee #1 for their feedback and will strengthen the hypotheses listed with additional details as well as modify the language to specify that seasonal variation is not formally investigated in this study.*

Line 80: unusual format for coordinates, add elevation of the KG hillslope, please adapt
*The authors will add the elevation of the KG hillslope to the text but keep the formatting of the coordinates consistent with the other studies associated with this campaign.*

Line 95ff, add species names of the dominant species of graminoids and dwarf shrubs.
*The authors will add species names as identified in associated studies by Salmon et al. (2009a-b).*

Line 101-112: rather unclear and wordy description. Be more precise here! Avoid expression such as initial phase and comprehensive informed phase (rather empty expressions).
*The authors will be more precise with their wording in this section.*

Line 114-135 unclear what means additional transect here, how many sampling points per transect?? Shorten! Please be more precise.
*The authors will shorten this section and clarify the number of sampling points per transect within the main text (information is currently outlined in Table S5 in the Supplement).*

Line 140ff: a nest of macro-rhizon: please define in the main text (not in the Supp. Material)- I wonder how long lasted the installation, for such short sampling intervals (of 4 days) the installation duration may affect the water sampling of the first day. Please explain!
*The authors will add these additional sampling details to the text. Rhizons were inserted in the nests on the first morning of each campaign and left in place for the duration of each campaign. Collection syringes were hung from the rhizons each morning and emptied each afternoon to obtain an integrated sample from each day in the field.*

Line 167. Unclear description, which transect (?) and soil 0-15cm has not been sampled?
*The authors will add these additional details of soil sampling to the Supplement in table format and reference appropriately within the main text. Yes, soil 0-15 cm was commonly dense with*

*roots from the overlying peat so we did not collect samples from 0-15 cm depth because we felt it would bias the chemistry. And all rhizon-obtained pore-water samples came from a depth of 15+ cm.*

Line 180: Why five litter samples when A1 and A4 have three sampling locations each, unclear…
*The authors were only able to obtain 5 uncompromised leaf samples. 6 samples were collected but one sample bag ripped and the sample was possibly compromised so we didn't analyze it, which is why only 5 samples were reported.*

Line 184: instead of each water sample…*In situ* parameters were measured for each water sampling location …
*The authors thank Anonymous Referee #1 and will adapt the text accordingly.*

The whole M & M section needs to strongly streamlined, now it is a potpourri of very different measurements and reader often does not know why for what purpose a measurement was carried out, besides when and how many times…I suggest to present all these different locations and campaigns in a Table.
*The authors thank Anonymous Referee #1 for this constructive suggestion and will present the different locations/campaigns in a Table and streamline the associated M&M text with appropriate references to the table. This suggestion will greatly improve the flow of the M&M section.*

Line 191-203 Statistical analysis: Weak description, no information how normal distribution, outliers, etc were handled. Weird description of processes acting on nitrate production.
*Additional details of the statistical analyses approach can be found in the Supplement. Authors will seek guidance from a statistical consultant for how to best display and communicate statistical outcomes obtained for next version.*

Line 205: I assume that soils (and correspondingly patches) along the slope differ from soils in the inundated lowlands. And permafrost occurrence and thickness of the layer- were they similar along the hillslope as in the lowlands? Please specify!
*The details of soil depth and soil moisture in the upland (UA + WA) and lowland (DA) sampling locations are included in Table S4 in the Supplement. The authors will add a statement identifying that lowland (DA) sites had shallower depths than upland (UA + WA) sites during Sept 2017 and July 2018, and add an appropriate reference to table S4 within the main text.*

Line 208: personal observation of whom? Mean gravimetric soil moisture content
*The personal observation was by those who participated in the field campaigns and measured active layer: ex: McCaully, Arendt, Newman, Heikoop, Wales, Musa. The authors will add this reference to the reference list to clarify. The authors will also alter current text 'mean soil moisture content (percent dry/wet weight)' to 'mean gravimetric soil moisture content' as suggested.*

Line 209: unclear: which other sampling campaigns?
*The authors will update the text here to specify that we are referring to the July 2017 and July 2018 sampling campaigns.*

Line 212: what are logistical and sampling challenges? Unclear.
*The authors feel that detailing the logistical challenges experienced over these campaigns would detract from the findings of the study but wanted to acknowledge that original plans had been disrupted and we were unable to get some of the data we had planned to get despite best intentions due to some logistical and sampling (field and equipment access) issues. However, the authors can also remove this statement from the text if it distracts from the manuscript.*

Line 213: adapt subtitle, 4 measuring days do not allow to delineate synoptic results (overstatement)
*The authors will modify the subtitle to "Initial results…" instead of "Synoptic results…".*

Line 214: you mean.. nitrate concentration was higher … please adapt

*The authors thank Anonymous Referee #1 and will adapt the text accordingly.*

Line 216: Strange description on atmospheric condition, air temperature, mean or maximum , please be precise, brief precipitation event, add where this has been measured and the exact rain amount.

*The authors will modify the text accordingly.*

Line 220: Seep? Not introduced in M& M.

*As commented on above, the authors will introduce the Seep during the M&M.*

Line 221: relative to the other three sampling days in July

*The authors will modify the text accordingly.*

Line 222: avoid such blue sky interpretation! Adapt

*The authors thank Anonymous Referee #1 for their caution and will adapt the associated text accordingly.*

Line 224: 58.61 mg L-1 Typing error, such high values are not presented in the Figures nor in Tables!

*The authors thank Anonymous Referee #1 for identifying this typo and will adapt the text accordingly.*

Line 230: please describe the weather conditions during your campaigns properly

*The authors thank Anonymous Referee #1 for this suggested and will add in more detailed daily weather information in a supplemental table., however, we were unable to obtain several quantitative weather parameters (ex: mm precip.) due to equipment access issues and equipment malfunction.*

Line 234-237: two times the same results with very different outcomes? Rather weird description, please improve.

*The authors will reword these lines to improve clarity.*

Line 228-258: Not fully clear what you like to present as results here, rather repetitive description on the different campaigns, not really convincing, please improve

*The authors will adapt the text to highlight the geochemical observations shown in Figures 2-6 and remove any repetitive language from the study designs.*

Line 259-271: Rather a potpourri of observation, in M &M section you mentioned 5 litter samples, now there are 10. What do these results tell you.

*The authors performed replicate measurements of the original 5 samples collected, making the total number of analyses points for the leaf litter, n = 10. The authors will add these details and clean up the text to streamline and clarify this section.*

Discussion: Obvious that the N2 fixation is the main source for the nitrate! So, please reorganise the whole discussion, now it reads like another result section! Please adapt

*The authors will take this feedback and use it to restructure the discussion section accordingly.*

Line 325-326: what kind of additional controls? Please avoid such empty sentences.

*The authors will specifically list examples of possible additional controls including denitrification bacteria, assimilation, and hydrologic flushing.*

Line: 329: A new aspect emerges: the comparison with the global meteoric water line! Why do you expect evaporation ins such a wet landscape? See line 244 Water flows even during periods without rain…

*The authors acknowledge the evaporation was negligible and made the comparison to the GMWL as a standard practice / common reference. Text cited from Line 244 was to active layer thaw*

*seepage, which is a constant presence of moisture but still susceptible to evaporation. The authors can remove the reference to the GMWL if readers find it to be distracting.*

Line 386: Future research? Rather bizzare that the authors list all the requirements for a more solid study. I would not declare these points as future research but as prerequisites for the current study!

*The authors argue that inviting the community to participate in more research in this area with suggestions to improve research outcomes is not bizarre, it is collegial and asks for engagement from the community. The goal of this text is to call on the community to increase our collective understanding of the ways in which NO3-N availability and mobility will change with changing climate in a variety of landscapes and environments. Knowing what we now know, we would have made some modifications to this study design and want to share these insights with other studies to they can build off of what is already known and maximize the outcomes of their research. Despite the challenges and learning opportunities encountered during this study, we still obtained data that increases our understanding of the highly variable nature and limited mobility of NO3-N in permafrost landscapes with alder shrub communities despite the increasing availability of NO3-N with warming climates and expanding alder stands.*

Line 401: Really unclear how such a single hillslope study should be of value for ESM

*This research was completed with oversight and support of the Next-Generation Ecosystems Experiments (NGEE Arctic) project. NGEE Arctic is supported by the Office of Biological and Environmental Research in the U.S. Department of Energy – Office of Science. A major motivation for this program is the use of modeling outcomes to inform experimental design and the use of experimental observations to inform models. We obtained experimental observations during this study that were shared with the modeling team to help inform the project on the appropriateness of incorporating small-scale spatial and temporal variations into models. While a single hillslope may not be incorporated into a larger ESM, processes observed within that hillslope can inform models of the general geochemical trends expected from expanding alder shrubs on Arctic hillslopes.*

Line 4902: bolster??

*The authors will replace 'bolster' with 'increase'.*

Conclusion: Already established that alder fix atmospheric N and therefore contribute to higher nitrate concentration in soil water. Rather redundant conclusions.

*The authors will critically assess their conclusions and remove instances of redundancy for future considerations. Authors also argue that additional studies that support known relationships add to n and increase the scientific community's broader understanding on nuanced variability within this known relationship.*

**Figures**

Figure1 (a) better to insert a map

*The authors will modify Figure 1a accordingly.*

Figure 2 a, b: Redundant to use different size of symbols and different colours. Please use for the same nitrate concentration the same colour! It is not really convincing to present means AND single values

*The authors find that although different size symbols and different colors for different NO3-N ranges may be redundant, it is useful to visually emphasize where the higher concentrations exist. The color and range of concentrations between Figures 1a and 1b will be verified for consistency.*

Figure 3: add sd (of bars) and add weather conditions (at least air, soil temperature and rain in mm)

*The authors included the sd information in Table S5 in the Supplement but can also add markers to Figure 3. The authors do not have all the weather condition information suggested but can add*

*air and soil temperature and the presence or absence of precipitation.*

Figure 4: There are no alders in the lower part of both transects, correct? I have some doubts whether the log Y scale really helps here. Table with the values (mean ± sd, number of replicates would be for sure more informative)

*There are no dense alder stands in the lowland sampling locations but there are small interspersed alders present (defined by alder savanna landscape). Tables S5 and S6 in the Supplement contain this requested information. The authors will include a reference to these tables in the Figure 4 caption.*

Figure 6. Boxplots largely overlap, that means no significant differences. How did you get this P<0.05? Though single t-tests? (Multiple mean test would be correct). Add n here…

*Table S7 in the Supplement contains n, max, min, mean, sd, and p-values for the $\delta^{18}O$ data portrayed in Figure 6. Authors may end up moving isotopic portion of this manuscript to the Supplement because it seems to distract from transportation outcomes and is the only part of the manuscript to focus on temporal variability so it may be better to separate the isotopic work including this figure from the main body of the manuscript.*

---

## Author Comment (AC2)

**Author response to Anonymous Referee #2 comments: High Temporal and Spatial Nitrate Variability on an Alaskan Hillslope Dominated by Alder Shrubs.**

*All author responses appear in grey italics below specific comments from Anonymous Referee #2.*
* * *
**General comments**

This study presents a comprehensive dataset which illustrates how substrate source (alder litter) and spatial connectivity in a sloping permafrost landscape may be larger control on $NO_3^-$ presence in uplands than soil moisture content, how increased $NO_3^-$ related to N fixer presence may be mobile in the landscape, and how redox conditions related to soil moisture and topography impacts the spatial extent of this mobility. This is a valuable contribution which underlines the importance of considering topography and N fixers as plant functional type in predictions of future plant N availability and potential $N_2O$ emissions.

However, this is a complicated dataset, and the study could benefit from a more coherent storyline, where the different datasets are presented not only in sequence, but are used together to tell a common story and the reader understands why the methods were chosen. This is done nicely in the abstract, but lacks in the discussion, where the $\delta^{18}O$ results and the $\delta^{15}N$ and $NO_3^-$ concentration results are, I suspect, not intended to be two separate stories, but they appear as such at the moment.

*The authors thank Anonymous Referee #2 for acknowledging the contribution of this study and for their constructive feedback. The authors have made note of the 'two separate stories' comment and will consider moving the isotopic data from the main text to the Supplement since it may distract from the main findings of our study. If we decide to keep the isotopic work in the main text, the authors will thoroughly rework the discussion to ensure the storyline is cohesive.*

**A few more specific section comments:**

Introduction: The language could use an overhaul, mainly a condensation of the text, where some points are repeated and some sentences/sections come out of context (see specific comments below).
*The authors thank Anonymous Referee #2 for their insights and will spend time rewording the introduction to condense the text and streamline our message.*

Materials and methods: The sample design is very comprehensive and complicated and as such benefits from a detailed description. However, the information could be more closely related to Figure 1b and 1c for clarity and condensed. The description of isotopic calculations is clear and useful. There is a lack of a quantitative estimate of precipitation (now currently addressed simply as "Precipitation events") from e.g. a micrometeorological station, as the precipitation downslope movement of $NO_3^-$ is such a central part of the results.
*The authors thank Anonymous Referee #2 for their insights and agree that the text should be linked more closely to Figure 1b and c for a visual reference to our design and will modify the M&M text accordingly. The authors also acknowledge that the precipitation events are lacking detail as these are based on in-person observations in the field and we unfortunately did not have the equipment to quantify precipitation adequately. However, not including these qualitative in-field observations of precipitation that correlate strongly to observed NO3-N transportation downslope along our transects, would be a disservice to the audience of this manuscript. So although we are left with qualitative rather than quantitative information for precipitation events, we choose to leave this information in the text.*

Discussion: The storyline of the discussion is not clear and appears more as a list of results related to literature

than a use of results to illuminate your research questions. As an example, the discussion of $\delta^{18}O$ related to precipitation events (line 328-339) comes a bit disconnected from the $NO_3^-$-story, but I suspect there is a point related to N transport, which needs to be clarified. The discussion needs to be restructured and condensed to tell the study story based on the results.

*The authors thank Anonymous Referee #2 for their insights and agree that the discussion could be strengthened to better emphasize the connections between our geochemical observations and N transport. The authors will rewrite this section for improved clarity with a streamlined storyline.*

Because of the large revisions needed in the communications of the results, I recommend that the manuscript can be reconsidered after major revisions.

*The authors thank Anonymous Referee #2 for their suggestion and will work to improve the quality of the writing to streamline our findings and strengthen emphasis on our outcomes.*

**Specific comments with line numbering**
Line 35-37: Which links and why is it important? Give one or two examples for a more engaging story.

*The authors thank Anonymous Referee #2 for this suggestion and will add examples of the linkages between C and N in the text, including "both C and N are of critical biological importance in Arctic coastal waters and changing fluxes from Arctic runoff are influencing primary productivity (McClelland et al., 2007), C to N ratios in Arctic tundra determine response of plant productivity to climate change (Weintraub et al., 2003), and that N likely exerts control on whether C is biogeochemically transformed in reactive catchments, which directly influences carbon-climate feedbacks (Koch et al., 2013).".*

Line 51-56: This second half of the paragraph seems a bit out of place, because the text introduces alder effects on soil chemistry above and continues below. Consider moving it and even skipping line 51-52 or replacing the sentence in line 39-40 as they say much the same.

*The authors thank Anonymous Referee #2 for their insights and will remove lines 51-52 and reword lines 53-36 to better align with the first half of the paragraph.*

Line 68: Alternatively "situated in a hillslope landscape"

*The authors will reword the current phrasing in the text "on a hillslope landscape" to this better suited suggestion.*

Line 91-91: I don't understand the function of this sentence in relation to the next sentences.

*The authors will reword the sentence to clarify that the alder shrub communities situated on the steep hillslopes exist in exclusively alder patches, whereas, in the alder savannas that exist on lowland water tracks, alders exist but are interspersed by other shrub communities.*

Line 160: A sentence on how $\delta^{18}O$ from $H_2O$ (soil solution) in your $NO_3^-$ is derived would be useful here.

*The authors thank Anonymous Referee #2 for this suggestion and will add in a sentence to clarify where $\delta^{18}O$ from $H_2O$ came from: "Isotopic data for $\delta^{18}O$ - $H_2O$ was measured directly from soil pore water samples using a GV Instruments Multiflow peripheral instrument (Heikoop et al., 2015)."*

186-190: Iron, sulfate and Manganese enter the story a bit abruptly here. If they have a function in the study design (as it is later clear that they have), please add a sentence earlier when explaining the study scope and strategy, adding the function of measuring those parameters.

*The authors thank Anonymous Referee #2 for bringing this to our attention and will add in a sentence to section 2.2 Sample Design to explain the importance of these supporting redox sensitive elements to our experiment earlier in the text.*

Line 269: You define all the other pools, but SON is not defined (Soil Organic Nitrogen, I assume)?

Line 284: This is an interesting finding from this study

*The authors thank Anonymous Referee #2 for noting this and will add additional text to the conclusions section to circle back to this point.*

Line 290-91: This statement, referring to Boshers et al. (2019) should be explained further. While the equation 1 is nicely explained previously, the argumentation for choosing this method should be discussed in relation to alternatives. You mention that "both possibilities" (line 291-292) are shown in figure 5. By this, I take that you mean the $H_2O$-derived only and the Eq. 1 determined predictions (?), however, I see only one interval of predictions in figure 5. The text and the link needs a better explanation.

*The authors thank Anonymous Referee #2 for this comment. Figure 5 should contain an additional predicted $\delta^{18}O$ range. The current range displayed is based on calculations from Kendall and McDonnell (1988) and a second should be added to show the predicted range of $\delta^{18}O$ based on calculations from Boshers et al., (2019). The authors will add in the additional predicted $\delta^{18}O$ range line for Boshers et al., (2019) to Figure 5 and better connect this figure to the text.*

Lines 328-339: This section is interesting and coherent in its argumentation, but its place in the story of the manuscript is not clear. The point may be that there is a connection to the $NO_3^-$ transport and –source, however, this link needs to be clearer for this section to be relevant to the overall story.

*The authors thank Anonymous Referee #2 for this insight and are considering moving the isotopic story to the Supplement associated with this manuscript because it seems to detract from our primary findings and acts only as support to show that we investigated beyond just NO3 transport: we also gained insights to the NO3-N sources present at our field site but were unable to find a clear linkage between transport and source. Thus, these findings may distract from our transport findings more than they add to it so it may be most appropriate to briefly acknowledge that this data exists but we were unable to find a clear connection to transport and readers can reference data if interested in the Supplement.*

Line 375-385: This section is a good example of clear, well-written communication/discussion of the results. !

*The authors thank Anonymous Referee #2 for highlighting this section and will use it as a reference to improve and streamline other sections.*

**Technical comments with line numbering:**

Line 19: The parentheses around $NO_3^-$ concentrations are not necessary and should either be removed or the sentence restructured

*The authors will restructure this sentence for improved flow and clarity.*

Line 32: Consider using "near-surface hydrologic conditions" in order to exclude e.g. subpermafrost groundwater

*The authors thank Anonymous Referee #2 and will change the language from 'hydrologic conditions' to 'near-surface hydrologic conditions' as aptly recommended.*

Line 181: a comma is likely missing between "bags" and "frozen".

*The authors thank Anonymous Referee #2 for catching this typo and will add a comma.*

Line 184: Soil temperature at which depth?

*The authors took soil sample measurements from the depth at which each rhizon was inserted, which*

*was 15 cm for the majority of sampling locations. The authors will add clarifying details to this line of text.*

Line 185: Introduce DO as Dissolved Oxygen before abbreviating

*The authors thank Anonymous Referee #2 for pointing out this oversight and will identify DO as Dissolved Oxygen before using the abbreviation.*

Line 200-201: Back up this statement with a reference?

*The authors will add the following references to at the end of this statement: O'Donnell and Jones, 2006; Moatar et al., 2017.*

*O'Donnell JA, and JB Jones. 2006. Nitrogen retention in the riparian zone of catchments underlain by discontinuous permafrost. Freshwater Biology 51: 854-856.*

*Moatar F, Abbot BW, Minaudo C, Curie, F, Pinay G. 2017. Elemental properties, hydrology, and biology interact to shape concentration-discharge curves for carbon, nutrients, sediment, and major ions. Water Resources Research 53: 1270-1287.*

Line 214: The beginning of this sentence should be reformulated – for once, the comma seems misplaced before "2017"

*The authors thank Anonymous Referee #2 and will reformulate this sentence for improved clarity.*

Figure 4: the lower part of the figure is cut off by the caption

*The authors thank Anonymous Referee #2 and will ensure the caption does not cut off the figure for future submissions.*

---

## Author Response (AR1)

**Author response to Anonymous Referee #1 comments:**

*All author responses appear in grey italics below specific comments from Anonymous Referee #1.*

The topic of the study to investigate the N cycle, in particular shifts in the nitrate concentration in the soil pore water in permafrost soils, affected by the $N_2$ fixation of alder (*Alnus viridis* ssp. *fruticosa*) is generally interesting. However, the study and the structure of the article has some major flaws and is a listing of observations instead of presenting a study with a clear focus. The structure of the article may be improved, but -unfortunately- several limitations of the design of the study cannot be corrected anymore (in retrospective).

*The authors thank Anonymous Referee #1 for their feedback and agree that there are some limitations of the data presented due to the disparate nature of field access and sampling campaigns. While a continuous time-series would have been preferable, we were limited due to scheduling permissions and availability but firmly believe there is inherit value in the data that we were able to collect and the following text details our motivation for the various campaigns. Since there had been no actual measurements of nitrate concentrations in soil pore water of Arctic alder stands at the time this investigation began, we conducted an initial reconnaissance study in year 1 to gain insights to the distributions and concentrations of pore water nitrate. We augmented these data with additional targeted sampling campaigns in the following year but were limited on timeframes of the follow-up investigations.*

**Aspects to can be improved:**

Many aspects that emerge in the result section, are not adequately described in the M&M section. Some descriptions given in the Supplementary Material

- "Seep" has not been explained in M&M

  *The authors thank Anonymous Referee #1 for this suggestion. The seep was initially defined in the Figure 3 caption but have added the following definition in the main text in section 2.1 for improved clarity. 'The A1 transect includes a sampling location we identify as a 'seep', which is a direct seep from the ground located at the slope transition between upland and lowland zones. The volume of water sourced from this seep was too small to measure directly but is estimated to be $< 2 \ cm^3 \ s^{-1}$. However, water actively trickled from this seep location during all sampling campaigns and is likely representative of active layer melt that surfaced at the upland-lowland transition.'*

- Patch of Alnus shrubland: dimension/ size of the patch not defined. Valuable pieces of information would be height of bushes, density of alder branches per m2. Please add.

  *The authors thank Anonymous Referee #1 for this suggestion. Figure 1c visually shows the scale of the alder stands we investigate and the respective sizes of A1 ($\sim 3400 \ m^2$) and A4 ($\sim 6400 \ m^2$) were already in the main text section 2.2. Salmon et al. (2019a) has published detailed vegetation parameters in a co-located study that we have added in as a reference in this section to direct readers who are interested in these metrics.*

- Authors present the total numbers of measuring points or samples (including soil pits), but it remains unclear what is really a replicate; a transect contains how many measuring points? The structure of the measuring points is also key for the statistical analysis. Both, temporal and spatial variability has to been taken into account in the statistical analysis. Here, one gains the impression, that one and then the other has lumped together. To emphasize that non parametric versus t-tests have been used is not adequate here.

  *The authors thank Anonymous Referee #1 for this suggestion and believe that the clarifying details we added to the text provide the needed context for interpretation of the results.*

  *At present, Figure 4 visually represents the relative locations and number of distinct rhizon clusters along the A1 and A4 transects, this information can be found in more detail in Table S5 in the Supplement, which identifies each specific sampling location that is clustered within the different sampling regions (i.e. A1_WI_Up, A1_WI_Mid, and A1_WI_Down, represent three distinct sampling locations that are located within Alder Stand 1 (A1)).*

*Table S6 in the Supplement displays the setup and results from the Mann Whitney rank sum tests performed for each constituent. The intention of the Mann Whitney statistical analyses is to look at the difference between 'upland' and 'lowland' sampling locations (spatial differences) and not between seasons (not temporal differences). Thus, here 'n' is the total number of samples collected in each location (upland and lowland) from both July and September 2018 combined. Based on advisement from a statistician consult, this approach appeared to be the best way to directly compare differences in 'upland' and 'lowland' sampling sites overall. The individual rhizon nests that make up 'upland' and 'lowland' are defined in Table S5. Text has been added to section 2.6 to communicate these distinctions and provide needed context to the reader.*

*We have modified and added text to the main manuscript to detail that the Mann Whitney statistical tests performed on nutrient data were not intended to highlight temporal variation, but to identify significant differences between the upland and lowland site geochemistry regardless of season.*

*Because of the discontinuous nature of our sampling campaigns, our data is not suited for temporal comparisons of nitrate but we were interested in the influence of seasonality and precipitation events on water isotopic signatures. Thus, the only temporal statistical analysis performed was on the water isotopes (Table S7), which looked at differences between each season and the overall mean of $\delta^{18}O$ of water (not nitrate). The total number of samples collected within each campaign define 'n' for each campaign (samples not grouped spatially). These samples are not divided into 'upland' and 'lowland' samples but are lumped holistically to show seasonal trends as opposed to spatial trends. The 'n' varies widely between months because the number of days per campaign varied. Because of the disparate temporal nature of our data, we have moved the analysis of temporal variation to the Supplemental material to allow the main text to focus on spatial variability and not detract from those findings.*

- The term "Alnus savanna" simply does not exist and is misguiding, please remove this term throughout the whole text!

  *The authors thank Anonymous Referee #1 for this comment but note that we use the term 'alder savanna', not 'alnus savanna'. As such, we respectfully disagree with removing this language as we can provide instances of 'alder savanna' terminology usage in both highly-referenced and recent literature and have added the following text to the manuscript to clarify any confusion that arises from use of this term for our audience.*

  *The authors have added the following text to the manuscript to provide context to the term: 'The term 'alder savanna' was defined by Frost et al., (2013), who identified 'Such shrubland communities, colloquially referred to as 'alder savannas' (Frost et al., 2013), have been described at several locations in Low Arctic, interior montane Alaska (Racine 1976, Racine and Anderson 1979, Chapin et al., 1989; Salmon et al., 2019a; Sulman et al., 2021). Regular spacing of alders in 'alder savannas' has been attributed to intra-specific competition for limiting nutrients (Chapin et al., 1989).''*

  ***The following references have been added to the manuscript:***
  *Chapin, F.S., McGraw, J. B., and Shaver, G. R. Competition causes regular spacing of alder in Alaskan shrub tundra. Oecologia 79, 412–416, https://doi.org/10.1007/BF00384322, 1989.*

  *Frost, G. V., Epstein, H. E., Walker, D. A., Matyshak, G., and Ermokhina, K. Patterned-ground facilitates shrub expansion in Low Arctic tundra. Environ. Res. Lett. 8, 015035. https://doi.org/10.1088/1748-9326/8/1/015035, 2013.*

  *Racine, C. H. Flora and vegetation: Biological Survey of the Proposed Kobuk Valley National Monument. Final Report ed H. R. Melchior (Fairbanks, AK: Alaska Cooperative Park Studies Unit, Biology and Resource Management Program, University of Alaska) pp 39–*

139, 1976.

Racine, C. H. and Anderson, J. H. *Flora and vegetation of the Chukchi-Imuruk area Biological Survey of the Bering Land Bridge National Monument: Revised Final Report ed H. R. Melchior (Fairbanks, AK: Alaska Cooperative Park Studies Unit, Biology and Resources Management Program, University of Alaska) pp 38–113, 1979.*

Sulman, B. N., Salmon, V. G., Iversen, C. M., Breen, A. L., Yuan, F., and Thornton, P. E. *Integrating Arctic plant functional types in a land surface model using above- and belowground field observations. J. Adv. Model. 13, e2020MS002396, https://doi.org/10.1029/2020MS002396, 2021.*

**Flaws in the design (cannot be correct anymore)**

- The nitrate concentrations in soil pore water of Alnus shrubland along the hill is compared to those in the soil pore water in the lowland. However, authors stated that the soil was partly covered by standing water and the alder bushes were much smaller in the lowland area or do not even exist in the lowland (Figure 4!) So, standing water on one hand may dilute the nitrate concentrations and create denitrifying conditions, and on the other hand smaller Alnus shrubs have for sure a lower $N_2$ fixation. These two aspects cannot be disentangled. So, it is obvious that the nitrate concentration is lower under such conditions, described here as major result (please avoid overstatements in general). Furthermore, it has already been shown that the nitrate concentrations in the soil water under Alnus stands is much higher than under non-Alnus stands due to the $N_2$ fixation (not a new result!)

  *The authors thank Anonymous Referee #1 for this comment and have modified the field location descriptions to clarify and accurately represent the conditions present in the uplands versus the lowland sampling sites. The lowland sampling sites are comprised of alder savanna, where alder are present but not in dense stands as they are in the uplands. The findings observed provide insight to the transport and mobility potential of nitrate from dense alder stands in upland environments: topography/gradient and precipitation/moisture in tundra environment with alders largely control this potential. So although nitrate availability may increase directly in the soil-pore water of dense alder stands, the moisture/precipitation and topography/gradient conditions of the alder environment likely limit/control of the mobility potential of this critical nutrient to downslope environments that we already know do not produce as much nitrate because of redox conditions and lesser alder density. The authors believe we have now properly clarified this point and have critically assessed the major results discussion and removed instances that could be interpreted as overstating outcomes.*

- On site weather station is fully missing. Although authors stated precipitation events as key for leaching processes.

  *A weather station has been installed for continuing studies at this NGEE field location but was unavailable at the time of the study presented here. We agree that measurements provided by a weather station would have been useful but are unable to retroactively obtain this information. Weather records of towns ~50 km away from the field site are accessible and referenceable but not a direct representation of the weather conditions at our remote field site and there are differences between these records and our recorded observations so we know they are not truly representative of our field conditions. Field observations of precipitation events are included for additional context of trends and variability observed. While we are unfortunately not able to quantify the extent of precipitation events, we are able to identify trends correlating with observed precipitation events that provide insight to nutrient transportation responses to weather observed.*

- A measuring campaign during 4 days (year 2017) is simply not representative for a seasonal measure (overstatement).

  *The authors agree with Anonymous Referee #1 and have modified the language accordingly to avoid temporal overstatements. While we do not have a full seasonal time series, the multiple*

*short time-series we have captured provide valuable insights to snapshots of compositions and variations that may exist within short timeframes within specific seasons.*

**Detailed comments:**

Abstract: Line 11: in the Arctic with capital letter, but arctic ecosystems with small letter. Please adapt throughout the MS.

*The authors thank Anonymous Referee #1 for pointing this inconsistency out and have ensured all instances of 'Arctic' are capitalized.*

Line 13: Simply not true, please consult the literature and adapt

*The authors appreciate the intention behind this suggestion and have modified this text to be better received while still communicating that knowledge gaps in this research area exist and identify the intention of our work to contribute to better understanding N dynamics (form, availability, and transportation potential) in a permafrost hillslope landscape.*

Line 14: Edaphic controls for the nitrate concentrations has not been shown.

*The authors have deleted 'edaphic' from the text.*

Line 18: I do not agree that all the nitrate is produced by degradation of N-rich alder shrub organic matter…or depends on your definition of organic matter..Are the corolla structure of the alder roots/nodules where the N2 fixation takes place, organic matter (I think not!)? The majority of the nitrate is already released during the N2 fixation.

*The authors thank Anonymous Referee #1 for this comment but believe there is misinterpretation of our phrasing, which is accurate as intended. The isotope ranges we calculated were for values corresponding to nitrate from the microbial degradation of N-rich alder shrub organic matter. This statement is not claiming that other forms/sources of nitrate production do not exist, it is identifying that we used isotopic ranges linked to microbial degradation of alder organic matter (from Kendall and McDonnell, 1998) to identify if the nitrate-N observed at our field location was produced from these processes/sources. Independent of these semantics, the authors have moved the isotopic component of our work to the Supplemental online material refocus attention on our non-isotopic nitrate variability story resulting from our geochemical work, so the language associated with this comment is no longer in the main text.*

Line 19, etc.: In general, better to express the nitrate concentrations as nitrate-**N** (enhanced comparability to other N compounds, atmospheric N deposition)

*The authors thank Anonymous Referee #1 for this comment and have changed 'nitrate' to 'nitrate-N' and 'NO$_3$-' to 'NO$_3^-$-N' when discussing/referencing concentrations.*

Line 23: denitrification buffers nitrate mobility. Strange description! Nitrate is transformed into N2 (complete) or N2O (incomplete) denitrification. From an ecological point of view the production of N2O is worse than nitrate mobility! Please adapt.

*The authors are not claiming that the production of N$_2$O isn't ecologically important, we are stating that nitrate-N is unable to be transported if it is converted to other N-species. Text has been modified to read 'denitrification limits the mobility of NO$_3^-$-N by transforming it to other N-species' and has been moved to the Supplemental online material with our other isotopic text.*

Line 24. Nutrient production is a misguiding term. Through N2 fixation nitrogen as a nutrient gets available, then it is transformed or lost again through complete denitrification..please adapt.

*The authors have altered text from 'nutrient production' to 'nutrient availability' here and in the remaining text.*

Line 40: Nutrient availability instead of nutrient production

*The authors thank Anonymous Referee #1 for this suggestion and have deleted this sentence due to a comment from Anonymous Referee #2 but have changed the language from 'nutrient production' to 'nutrient availability' in several other instances in the text where that phrasing is more appropriate.*

Line 46: Latin names – italic

*The authors thank Anonymous Referee #1 and have changed the formatting in this instance accordingly and have verified that all Latin names are now in italics.*

Line 48: instead of microbes, add here: *Frankia* bacteria

*The authors thank Anonymous Referee #1 for this suggestion and have altered the text from 'microbes to 'Frankia bacteria' as suggested.*

Line 51, line 57: Wrong!! Brühlmann et al. 2014, Hiltbrunner et al. 2014: Both studies are located in the montane (not in the alpine vegetation belt of the Alps), but clearly not on permafrost soils! Increase of alder shrubland due to changes in land use, not increasing temperatures. Please correct and add this aspect of land use changes.

*The authors thank Anonymous Referee #1 for identifying this discrepancy and have modified the language in the original line 51 to refer to these studies as taking place in cold environments instead of permafrost environments. The authors have also modified the language associated with the reference in the original line 57 reference upslope/downslope nutrient mobility observed regardless of permafrost presence. We attempted to add text referencing land-use changes as a cause of shifting nitrate availability but found that because land use changes are not present in our study landscape, it introduced another factor that detracted from a streamlined introduction. Thus, we chose to modify the text associated with the references in question rather than introduce land use changes.*

Line 68-70: Necessary?

*The authors have deleted this text from the Study Objectives section because these details are already included in the Acknowledgements section.*

Line 73-76: Hypotheses rather weak as already widely known that alder shrubs through their N2 fixation are source for nitrate. And your measuring campaigns cover some days in July and September, not seasons. And see comments on line 18 and 23 (comments are not repeated here).

*The authors thank Anonymous Referee #1 for their feedback and believe we have strengthened the hypotheses listed with additional details and have modified the language to avoid any interpretations that seasonal variation was formally investigated in this study.*

Line 80: unusual format for coordinates, add elevation of the KG hillslope, please adapt

*The authors have edited the coordinate formatting to match that from Salmon et al. (2019a) and have added in the elevation range of the hillslope (40-140 m.a.s.l.).*

Line 95ff, add species names of the dominant species of graminoids and dwarf shrubs.

*The authors have added in examples of species names as identified in associated studies by Salmon et al. (2009a-b).*

Line 101-112: rather unclear and wordy description. Be more precise here! Avoid expression such as initial phase and comprehensive informed phase (rather empty expressions).

*The authors thank Anonymous Referee #1 for this comment and have modified and added to this section to more precisely describe our sampling location.*

Line 114-135 unclear what means additional transect here, how many sampling points per transect?? Shorten! Please be more precise.

*The authors have shortened this section and added references to the supplemental tables that detail the number of sampling points per transect per sampling campaign. The number of sampling locations per transect increased with each campaign to get more detailed spatial resolution of nitrate variability along the transects. The authors have chosen to keep these details in a table in the Supplement since they are extensive and distract from the take-aways points we convey in the main text. However, the authors have also have moved some text from the Supplemental material to the main text (in section 2.3) that provides context to the number of rhizons per nest and the timeframe that the rhizons were installed.*

Line 140ff: a nest of macro-rhizon: please define in the main text (not in the Supp. Material)- I wonder

how long lasted the installation, for such short sampling intervals (of 4 days) the installation duration may affect the water sampling of the first day. Please explain!

*The authors thank Anonymous Referee #1 for this suggestion and have added these additional sampling details to the text in section 2.3.*

Line 167. Unclear description, which transect (?) and soil 0-15cm has not been sampled?

*The authors have modified the language in this section to improve clarity.*

Line 180: Why five litter samples when A1 and A4 have three sampling locations each, unclear…

*The authors modified and added the following text to the manuscript to clarify, 'Alder leaf litter was collected in September 2018 from six locations along the A1 and A4 transects (3 samples from each transect), stored in sealed plastic bags frozen, and homogenized prior to analysis (n=6). A contamination issue occurred with one of the leaf litter samples collected from the A4 transect, leaving us with n = 5.*

Line 184: instead of each water sample…*In situ* parameters were measured for each water sampling location …

*The authors thank Anonymous Referee #1 and adapted the text from 'each water sample' to 'each water sampling location.'*

The whole M & M section needs to strongly streamlined, now it is a potpourri of very different measurements and reader often does not know why for what purpose a measurement was carried out, besides when and how many times…I suggest to present all these different locations and campaigns in a Table.

*The authors thank Anonymous Referee #1 for suggestion. These details were already included in Supplemental Table S4 but could have been referenced more effectively in the main text. The authors have moved the isotopic text to the Supplemental material to avoid distracting from the nitrate variability story. The authors have also streamlined the main M&M text with appropriate references to the relevant supplemental material and have identified the intended purpose of each parameter collected within the M&M subsections. These edits and added references have improved the flow of this section and provided additional context to the readers.*

Line 191-203 Statistical analysis: Weak description, no information how normal distribution, outliers, etc were handled. Weird description of processes acting on nitrate production.

*Authors have added in the requested details to the main text and additional details of the statistical analyses approach can be found in the Supplemental material. Authors met with a statistical consultant to verify that approach used was appropriate for the dataset obtained and have been reassured that the approach used is valid and appropriate given the nature of our data collected.*

Line 205: I assume that soils (and correspondingly patches) along the slope differ from soils in the inundated lowlands. And permafrost occurrence and thickness of the layer- were they similar along the hillslope as in the lowlands? Please specify!

*The details of soil depth and soil moisture in the upland (UA + WA) and lowland (DA) sampling locations are included in Table S4 in the Supplement and the authors have added sentences and modified language in the main text to clarify similarities/differences and direct the reader to the Supplemental Table (S4) that contains additional details. Text already existed that identified that the active layer depth was greater in the lowland portions of the transects than the upland portions of the transects for September 2017 and July 2018.*

Line 208: personal observation of whom? Mean gravimetric soil moisture content

*The personal observation was by those who participated in the field campaigns, who are identified in the 'Author Contributions' section. The authors have added clarifying text to this reference. The authors have also altered the text 'mean soil moisture content' to 'mean gravimetric soil moisture content' as suggested.*

Line 209: unclear: which other sampling campaigns?

*The authors have updated the text here from 'other' to 'the July 2017 and July 2018' for clarity.*

Line 212: what are logistical and sampling challenges? Unclear.

*The authors feel that detailing the logistical challenges experienced over these campaigns is unnecessary and would distract from the findings of the study but wanted to acknowledge that original plans had been disrupted and we were unable to get some of the data we had planned to get despite best intentions due to some logistical and sampling (field and equipment access) issues. The authors have chosen to leave the text as is.*

Line 213: adapt subtitle, 4 measuring days do not allow to delineate synoptic results (overstatement)

*The authors have modified the subtitle to 'initial results…' instead of 'synoptic results…'.*

Line 214: you mean.. nitrate concentration was higher … please adapt

*The authors thank Anonymous Referee #1 and have modified the text from 'was significantly greater' to 'concentrations were significantly higher'.*

Line 216: Strange description on atmospheric condition, air temperature, mean or maximum, please be precise, brief precipitation event, add where this has been measured and the exact rain amount.

*The authors have modified this text and have stated that a quantitative rain amount is unfortunately not available for this precipitation event.*

Line 220: Seep? Not introduced in M& M.

*As commented on above, the authors have introduced the Seep in section 2.1.*

Line 221: relative to the other three sampling days in July

*The authors have modified the text from 'all other sampling days' to 'the other sampling days from the initial July 2017 sampling campaign.'.*

Line 222: avoid such blue sky interpretation! Adapt

*The authors thank Anonymous Referee #1 for their caution but believe this explanation is the likely reason this seep exists and have phrased this sentence as such. We do not see a reason to overcomplicate an explanation that the field team agrees with after months of interacting with this location over several years.*

Line 224: 58.61 mg L-1 Typing error, such high values are not presented in the Figures nor in Tables!

*The authors thank Anonymous Referee #1 for identifying this typo and have adapted the text accordingly.*

Line 230: please describe the weather conditions during your campaigns properly

*The authors thank Anonymous Referee #1 for this suggestion and agree that this study would benefit from these details if they were available but we were unable to obtain several quantitative weather parameters (ex: mm precip.) due to equipment access issues and equipment malfunction.*

Line 234-237: two times the same results with very different outcomes? Rather weird description, please improve.

*The authors have reworded these lines and added details to improve clarity.*

Line 228-258: Not fully clear what you like to present as results here, rather repetitive description on the different campaigns, not really convincing, please improve

*The authors thank Anonymous Referee #1 for this feedback and have modified the text and removed any obvious campaign description repetition from the text but believe that this structure allows for the authors to communicate major observations/results from within each separate field campaign of Phase 2 of this study. However, we believe the modifications we made have improved this section even though the structure remains the same.*

Line 259-271: Rather a potpourri of observation, in M &M section you mentioned 5 litter samples, now there are 10. What do these results tell you.

*The authors have moved this section to the Supplemental material and added in clarifying details including that we performed replicate measurements of the original 5 samples collected, making the total number of analyses points for the leaf litter, n = 10.*

Discussion: Obvious that the N2 fixation is the main source for the nitrate! So, please reorganise the whole discussion, now it reads like another result section! Please adapt

*The authors thank Anonymous Referee #1 for this feedback, have moved the source/isotopic text to the Supplement material, and have restructured the discussion section accordingly.*

Line 325-326: what kind of additional controls? Please avoid such empty sentences.

*The authors have added examples of possible additional controls including denitrification bacteria, assimilation, and hydrologic flushing.*

Line: 329: A new aspect emerges: the comparison with the global meteoric water line! Why do you expect evaporation ins such a wet landscape? See line 244 Water flows even during periods without rain…

*The authors acknowledge the evaporation was negligible and made the comparison to the GMWL as standard practice / common reference. Text cited from Line 244 was to active layer thaw seepage, which is a constant presence of moisture but still susceptible to evaporation. The authors have removed the sentence with the reference to the GMWL to avoid distracting the reader.*

Line 386: Future research? Rather bizzare that the authors list all the requirements for a more solid study. I would not declare these points as future research but as prerequisites for the current study!

*The authors have modified the language in this section to acknowledge areas that future studies could build off of and not dwell on the perceived shortcomings of this study. The authors also respectfully argue that inviting the community to participate in more research in this area with suggestions to improve research outcomes is not bizarre, it is collegial and asks for engagement from the community. The goal of this text is to call on the community to increase our collective understanding of the ways in which $NO_3^-$ availability and mobility will change with changing climate in a variety of landscapes and environments. Knowing what we now know, we would have made some modifications to this study design and/or incorporated additional parameters and want to share these insights with other studies to they can build off of what is already known and maximize the outcomes of their research. Despite the challenges and learning opportunities encountered during this study, we still obtained data that increases our understanding of the highly variable nature and limited mobility of $NO_3^-$ in permafrost landscapes with alder shrub communities despite the increasing availability of $NO_3^-$ with warming climates and expanding alder stands.*

Line 401: Really unclear how such a single hillslope study should be of value for ESM

*This research was completed with oversight and support of the Next-Generation Ecosystems Experiments (NGEE Arctic) project. NGEE Arctic is supported by the Office of Biological and Environmental Research in the U.S. Department of Energy – Office of Science. Field and lab outcomes and observations collected under NGEE Arctic are used to inform Earth System Models (ESM) through the collection and incorporation of experimental data in the face of increasing Arctic temperatures. A major motivation for this program is the use of modeling outcomes to inform experimental design and the use of experimental observations to inform models. We obtained experimental observations during this study that were shared with the modeling team to help inform the project on the appropriateness of incorporating small-scale spatial and temporal variations into models. While a single hillslope may not be incorporated into a larger ESM, processes observed within that hillslope can inform models of the general geochemical trends expected from expanding alder shrubs on Arctic hillslopes.*

Line 4902: bolster??

*The authors have replaced 'bolster' with 'increase'.*

Conclusion: Already established that alder fix atmospheric N and therefore contribute to higher nitrate concentration in soil water. Rather redundant conclusions.

*The authors have removed this phrasing from our conclusions but are also of the mindset that publication of additional studies that support known relationships add to n and increase the scientific community's broader understanding on nuanced variability within this known relationship. Corroborating studies are useful especially when they take place in different*

*landscape or climate settings.*

**Figures**

Figure1 (a) better to insert a map
*The authors have modified Figure 1 accordingly.*

Figure 2 a, b: Redundant to use different size of symbols and different colours. Please use for the same nitrate concentration the same colour! It is not really convincing to present means AND single values
*The authors find that although different size symbols and different colors for different $NO_3^-$-N ranges may be redundant, it is useful to visually emphasize where the higher concentrations exist. The color and range of concentrations between Figures 2a and 1b have been verified for consistency.*

Figure 3: add sd (of bars) and add weather conditions (at least air, soil temperature and rain in mm)
*The authors had included the sd information in Table S5 in the Supplement but have also added sd markers to Figure 3. The authors are unable to add the requested weather conditions that correspond to our field site due to equipment malfunction but the figure contains asterisk that denote days on which precipitation events were observed by those involved in the field campaign. We wish we were able to quantify the requested weather parameters but have indicated which days precipitation occurred to provide further insights to our data interpretation.*

Figure 4: There are no alders in the lower part of both transects, correct? I have some doubts whether the log Y scale really helps here. Table with the values (mean ± sd, number of replicates would be for sure more informative)
*There are no dense alder stands in the lowland sampling locations but there are small interspersed alders present (defined by alder savanna landscape). However, our transect did not directly intersect with any alder shrubs in the lowland area. Tables S5 and S6 in the Supplement contain this requested information. The authors have included a reference to these tables in the Figure 4 caption. The authors have created this figure in both linear and log scale and prefer the log scale to emphasize differences in the measured redox concentrations.*

Figure 6. Boxplots largely overlap, that means no significant differences. How did you get this P<0.05? Though single t-tests? (Multiple mean test would be correct). Add n here…
*Table S7 in the Supplement contains n, max, min, mean, sd, and p-values for the $\delta^{18}O$ data portrayed in the original Figure 6. Authors moved the isotopic portion of this manuscript to the Supplement because it seems to distract from transportation outcomes and is the only part of the manuscript to focus on temporal variability so we separated the isotopic work including this figure from the main body of the manuscript.*
* * *
**Author response to Anonymous Referee #2 comments:**
*All author responses appear in grey italics below specific comments from Anonymous Referee #2.*

**General comments**

This study presents a comprehensive dataset which illustrates how substrate source (alder litter) and spatial connectivity in a sloping permafrost landscape may be larger control on $NO_3^-$ presence in uplands than soil moisture content, how increased $NO_3^-$ related to N fixer presence may be mobile in the landscape, and how redox conditions related to soil moisture and topography impacts the spatial extent of this mobility. This is a valuable contribution which underlines the importance of considering topography and N fixers as plant functional type in predictions of future plant N availability and potential $N_2O$ emissions.

However, this is a complicated dataset, and the study could benefit from a more coherent storyline, where the different datasets are presented not only in sequence, but are used together to tell a common story and the reader understands why the methods were chosen. This is done nicely in the abstract, but lacks in the discussion, where the $\delta^{18}O$ results and the $\delta^{15}N$ and $NO_3^-$ concentration results are, I suspect, not intended to be two separate stories, but they appear as such at the moment.

*The authors thank Anonymous Referee #2 for acknowledging the contribution of this study and for their constructive feedback. The authors have made note of the 'two separate stories' comment and have decided to move the isotopic data from the main text to the Supplement since it may distract from the main nitrate variability and mobility findings of our study. The authors have reworked the remaining discussion in the main text to ensure the storyline is cohesive.*

**A few more specific section comments:**

Introduction: The language could use an overhaul, mainly a condensation of the text, where some points are repeated and some sentences/sections come out of context (see specific comments below).
*The authors thank Anonymous Referee #2 for their insights and have substantially reworded and condensed the introduction to streamline our message.*

Materials and methods: The sample design is very comprehensive and complicated and as such benefits from a detailed description. However, the information could be more closely related to Figure 1b and 1c for clarity and condensed. The description of isotopic calculations is clear and useful. There is a lack of a quantitative estimate of precipitation (now currently addressed simply as "Precipitation events") from e.g. a micrometeorological station, as the precipitation downslope movement of $NO_3^-$ is such a central part of the results.
*The authors thank Anonymous Referee #2 for their insights and agree that the text should be linked more closely to Figure 1 for a visual reference to our design and have modify the M&M text to include additional references to Figure 1 accordingly. The authors also acknowledge that the precipitation events are lacking detail as these are based on in-person observations in the field and we unfortunately did not have the equipment to quantify precipitation adequately. However, not including these qualitative in-field observations of precipitation that correlate strongly to observed NO₃-N transportation downslope along our transects, would be a disservice to the audience of this manuscript. So, although we are left with qualitative rather than quantitative information for precipitation events, we choose to leave this information in the text.*

Discussion: The storyline of the discussion is not clear and appears more as a list of results related to literature than a use of results to illuminate your research questions. As an example, the discussion of $\delta^{18}O$ related to precipitation events (line 328-339) comes a bit disconnected from the $NO_3^-$-story, but I suspect there is a point related to N transport, which needs to be clarified. The discussion needs to be restructured and condensed to tell the study story based on the results.
*The authors thank Anonymous Referee #2 for their insights and have modified the discussion to better emphasize the connections between our geochemical observations and N transport. The authors have also decided to move the isotopic work to the Supplement to allow the main body of text to center around $NO_3^-$-N.*

Because of the large revisions needed in the communications of the results, I recommend that the manuscript can be reconsidered after major revisions.
*The authors thank Anonymous Referee #2 for their suggestion and have worked to improve the quality of the writing to streamline our findings and strengthen emphasis on our outcomes.*

**Specific comments with line numbering**
Line 35-37: Which links and why is it important? Give one or two examples for a more engaging story.
*The authors thank Anonymous Referee #2 for this suggestion but have ultimately decided to remove this sentence since we felt it detracted from the streamlined revised nature of our introduction after adding in examples of C and N links and their importance. The authors feel that focusing on N without the introduction of C is more appropriate for the goals of this manuscript.*

Line 51-56: This second half of the paragraph seems a bit out of place, because the text introduces alder effects on soil chemistry above and continues below. Consider moving it and even skipping line 51-52 or replacing the sentence in line 39-40 as they say much the same.

*The authors thank Anonymous Referee #2 for their insights have altered the text from original lines 39-40 to encompass the content from original lines 51-52 since it was repetitive. Original lines 53-56 have been moved to the previous paragraph to avoid breaking up the alder effects on soil chemistry text. This arrangement seems to transition between and address topics in a more fluid manner.*

Line 68: Alternatively "situated in a hillslope landscape"

*The authors have reworded the phrasing in the text from "on a hillslope landscape" to this better suited suggestion.*

Line 91-91: I don't understand the function of this sentence in relation to the next sentences.

*The authors thank Anonymous Referee #2 for this comment and have reworded this sentence and the following sentences for improved clarity. These sentences are intended to highlight the unique characteristics of an 'alder shrubland' designation versus an 'alder savanna' designation and we believe the edits made clarify these distinctions.*

Line 160: A sentence on how $\delta^{18}O$ from $H_2O$ (soil solution) in your $NO_3^-$ is derived would be useful here.

*The authors thank Anonymous Referee #2 for this suggestion and have added in a sentence to clarify where $\delta^{18}O$ from $H_2O$ came from: "Isotopic data for $\delta^{18}O$ - $H_2O$ was measured directly from soil pore water samples using a GV Instruments Multiflow peripheral instrument (Heikoop et al., 2015)." This text has been moved to the Supplemental material along with our other isotopic text and figures.*

186-190: Iron, sulfate and Manganese enter the story a bit abruptly here. If they have a function in the study design (as it is later clear that they have), please add a sentence earlier when explaining the study scope and strategy, adding the function of measuring those parameters.

*The authors thank Anonymous Referee #2 for bringing this to our attention and have added text to section 2.2 Sample Design to explain the importance of these supporting redox sensitive elements to our experiment earlier in the text.*

Line 269: You define all the other pools, but SON is not defined (Soil Organic Nitrogen, I assume)?

*The authors thank Anonymous Referee #2 for identifying this location of SON that would benefit from further clarification. Yes, SON is Soil Organic Nitrogen. The authors previously define SON in section 2.4 but have added in the full term here for consistency in the sentence structure and to avoid any confusion of the term.*

Line 284: This is an interesting finding from this study

*The authors thank Anonymous Referee #2 for noting this and have modified text within the conclusion section to circle back to this point.*

Line 290-91: This statement, referring to Boshers et al. (2019) should be explained further. While the equation 1 is nicely explained previously, the argumentation for choosing this method should be discussed in relation to alternatives. You mention that "both possibilities" (line 291-292) are shown in figure 5. By this, I take that you mean the $H_2O$-derived only and the Eq. 1 determined predictions (?), however, I see only one interval of predictions in figure 5. The text and the link needs a better explanation.

*The authors thank Anonymous Referee #2 for this comment and have added clarity to this text. Upon further investigation the authors determined that the range provided in this figure encapsulates both predictive ranges so instead of showing two smaller ranges, we combined the predictive ranges into one that covers both predictive ranges (from Kendall and McDonnell, 1988 and Boshers et al., 2019). This text has been moved to the Supplement online material along with the other isotopic work presented in this study to allow the main text to focus on the nitrate variability and mobility story.*

Lines 328-339: This section is interesting and coherent in its argumentation, but its place in the story of the manuscript is not clear. The point may be that there is a connection to the $NO_3^-$ transport and –source, however, this link needs to be clearer for this section to be relevant to the overall story.

*The authors thank Anonymous Referee #2 for this insight and have moved the isotopic story to the Supplement associated with this manuscript because it seems to detract from our primary findings and*

*acts only as support to show that we investigated beyond just NO$_3$ variability and transport: we also gained insights to the NO$_3^-$-N sources present at our field site but did not find a clear linkage between transport and source. Thus, these findings may distract from our transport findings more than they add to it so it may be most appropriate to briefly acknowledge that this data exists and readers can reference it in the Supplement if interested.*

Line 375-385: This section is a good example of clear, well-written communication/discussion of the results. !
*The authors thank Anonymous Referee #2 for highlighting this section and used it as a reference to improve and streamline other sections.*

**Technical comments with line numbering:**

Line 19: The parentheses around NO$_3^-$ concentrations are not necessary and should either be removed or the sentence restructured
*The authors thank Anonymous Referee #2 for this suggestion and have removed the parenthesis around the first instance of nitrate-N concentrations to improve the flow of the sentence.*

Line 32: Consider using "near-surface hydrologic conditions" in order to exclude e.g. subpermafrost groundwater
*The authors thank Anonymous Referee #2 and have changed the language from 'hydrologic conditions' to 'near-surface hydrologic conditions' as aptly recommended.*

Line 181: a comma is likely missing between "bags" and "frozen".
*The authors thank Anonymous Referee #2 for catching this typo and have added a comma.*

Line 184: Soil temperature at which depth?
*The authors took soil sample measurements from the depth at which each rhizon was inserted, which was 15 cm for the majority of sampling locations. The authors have added these clarifying details to this line of text.*

Line 185: Introduce DO as Dissolved Oxygen before abbreviating
*The authors thank Anonymous Referee #2 for pointing out this oversight and have added '(DO)' after the first reference to 'dissolved oxygen' in the previous sentence to clearly define this abbreviation.*

Line 200-201: Back up this statement with a reference?
*The authors have added the following references to at the end of this statement: O'Donnell and Jones, 2006; Moatar et al., 2017, and have added the Moatar et al., 2017 reference to the reference list.*

*O'Donnell JA, and JB Jones. 2006. Nitrogen retention in the riparian zone of catchments underlain by discontinuous permafrost. Freshwater Biology 51: 854-856.*

*Moatar F, Abbot BW, Minaudo C, Curie, F, Pinay G. 2017. Elemental properties, hydrology, and biology interact to shape concentration-discharge curves for carbon, nutrients, sediment, and major ions. Water Resources Research 53: 1270-1287.*

Line 214: The beginning of this sentence should be reformulated – for once, the comma seems misplaced before "2017"
*The authors thank Anonymous Referee #2 and have removed the comma and reformulated this sentence for improved clarity.*

Figure 4: the lower part of the figure is cut off by the caption
*The authors thank Anonymous Referee #2 and have edited the placement of Figure 4 and the corresponding caption to ensure the full figure is visible in this version of our submission.*

---

## Referee Report (RR1)

**Second revision of manuscript tc-2021-166:**

**Title: High Nitrate Variability on an Alaskan Permafrost Hillslope Dominated by Alder Shrubs**

**Authors: R. E. McCaully, C.A. Arendt, B. D. Newman1, V. G. Salmon3, J. M. Heikoop, C. J. Wilson, S. A. Sevanto, N. A. Wales, G. B. Perkins, O. C. Marina, and S. D. Wullschl.**

**Comments to the manuscript (second revision)**
This manuscript has radically improved during revision and now presents a much clearer storyline and has a logical link from results to the conclusions, which are presented much more clearly. The limits to the study that cannot be changed are still there, but are acknowledged and conclusions can be made with that in mind.I have a few specific comments and otherwise recommend acceptance of the manuscript.

Abstract
Nice and clear writing here

Introduction:
I appreciate how the  authors have improved the storyline –  the build up is much better now.

Materials and methods
I like the additional details on the statistics. It is a complicated study and thus needs a long material and methods section.

Results and discussion
The story here is much improved and the summary of the results of the isotopic data functions well as a comment that adds nuance to the main story.
Interpretation of the data is now in a coherent way arguing for the main conclusions of the study, and the isotopic data is used well to support your argument.

Conclusion
Nice summary of your conclusions, why you ended there and why they matter

**Specific comments with line numbering**
L 124: Is there a word missing between 'down' and 'gradient' ?
L 126: Nice with the description of why you measured Mn and Fe  :)
L 145: 'hereafter' in stead of 'hereby'?
L 266: the (21-22) refers to the date, I assume – maybe change to (21-22nd) or (21-22.)
L 283: Why is the labile N inaccessible according to Darouset-Nardi and Weintraub (2014)?
L 291: If you want a more recent reference, Rasmussen et al. 2020 also saw a flush of organic C and N during and after a rainfall event.
L 335-337: Agreed. And assessing the time around snowmelt and soil thaw and the transport potential related to redox environments present there is also necessary in the future.

Supplementary: A few of the references are underlined

Refs.

Rasmussen, LH, Michelsen, A, Ladegaard-Pedersen, P., Skov Nielsen, C. and Elberling, B. (2020). Arctic soil water chemistry in dry and wet tundra subject to snow addition,
summer warming and herbivory simulation Soil Biology and Biochemistry 141 (2020) 107676.

---

## Author Response (AR2)

**Author Responses to Anonymous Referees:**

*All author responses appear in grey italics below specific comments from Anonymous Referees. The authors thank all referees for the time and energy they have spent providing constructive feedback on this body of work. We have summarized our responses below, have responded to each comment individually on the following pages, and we believe we have adequately addressed the feedback provided.*

*Previous reviewer comments have helped us greatly improve the readability of this manuscript through the reorganization of the main text and supplement, which focuses the main text on variability trends observed and moves the additional supporting work to the Supplement to ensure readers have access to all work associated with this study if interested.*

*The co-occurrence of precipitation events on days where enhanced nitrate availability occurred, must be recognized, even without quantitative precipitation data and it would be irresponsible for the authors to ignore the precipitation events observed in the field.*

*While we appreciate that Anonymous Referee #1 has stylistic preferences and respect their opinion, Anonymous Referees #2-3 do not take issue with these same areas of concern. Anonymous Referee #1's lack of specific details made it challenging to understand/address their comments. For example, what in the abstract is not discussed within the text? This is not specified, nor why the reviewer considers the abstract information not new. Is the reviewer aware of other studies that have compared pore water nitrate concentrations in Alaskan Alder permafrost systems? If so, we would be glad to cite them. However, there do not appear to be any that exist. So, how can this work not be new? It is unfounded to claim a lack of new contributions because Alder nitrate studies exist within other landscapes when Alaskan Arctic systems have had so little investigation and the data and findings presented here are original to this study and provide insights to these understudied systems.*

*Furthermore, Anonymous Referee #1 did not believe this study should be published partially based on design, but reconnaissance studies are valuable to the scientific community, and this study contributes original data from a region that does not have alder influenced soil pore water nitrate variability published previously. Anonymous Referees #2-3 were able to see the merit in our reconnaissance study outcomes and have provided constructive feedback that has allowed this body of work to improve. The authors firmly believe that comments from Anonymous Referee #1 serve as unnecessary gatekeeping and that our study provides original data and important insights to the scientific community that are worthy of publication.*
* * *
**Anonymous Referee #1:**

I see that the authors tried to restructure the text, but the MS in the presented state cannot be published, sorry.

The structure is still not satisfying: several repetitions (on flushing by rainfall), part of the discussion is presented like result section, etc.

Because the whole MS needs to be restructured again, I will not give here a specific line- per-line comment. And please (a detail), do not reply to reviewers by using the same sentence/expressions all the time, this is more than boring.

*The authors thank Anonymous Referee #1 for their time and input but respectfully and strongly disagree on final recommendation and stylistic notes of Anonymous Referee #1. We can appreciate the frustration*

*of Anonymous Referee #1 and understand that they would have designed this study differently. However, just because this is a reconnaissance study (which helps form the basis of more detailed experimental designs at our site and elsewhere) does not mean this work is unworthy of publication. The reviewer has lost sight of the fact that these data are a unique new data set that benefits scientific understanding in Arctic ecosystems that are changing rapidly. What benefit is served by denying the scientific community access to the data and findings presented in this study, because the reviewer simply doesn't like the design? The data are valid and insightful, and this study certainly serves as a useful baseline for future Arctic studies on Alder impacts and these results are important to communicate to the broader scientific community.*

Here, my main reason why the MS cannot be accepted for publication:

1) There is a general and deep misunderstanding, an article needs to be written and condensed to a unit, important pieces of information have to be incorporated in the article and not presented in the Supplementary Material or in a co-located study or somewhere (by the way, this was stated already in the first review). You cannot formulate statements in the abstract and the data are fully absent in the article. The reader of your article MUST find your "main story" in the MS, not in the Supplementary Material.

*This comment cannot be sufficient as the main reason that Anonymous Referee # 1 recommends the article be rejected as the comment indicates. It is simply based on a stylistic difference of opinion. Rejection of a paper should not be based mainly on what the reviewer "prefers" when all data and cited information are available for anyone who is interested, and the main points of the paper are easily understood within the main text. We find this recommendation is inappropriate given that it is based on one reviewer's rather narrow idea of how to formulate a manuscript, and because the other reviewers appreciate the edits made that make the text compact and to the point. The authors have significantly condensed the text to follow a common theme surrounding variability and all data central to the main text is included within the main text. Data outside the scope of the findings in our main body of text have been shifted to the Supplement so that if a reader is interested in additional parameters measured, they have access to that data for additional context, but it does not distract from our main story. We believe that the transition of the nitrate source (isotopic) content to the Supplement has greatly benefitted this body of work to highlight the degree of nitrate variability observed more than any other study outcome. The authors also argue that the abstract does not present anything that is not included within the main body of text and are unsure as to what this comment refers to.*

2) The data presented as main result in the abstract, are simply not new:

"Soil pore water collected within alder shrubland had an average NO3--N (nitrogen from nitrate) concentration of 4.27 ± 8.02 mg L-1 and differed significantly from locations outside alder shrubland (0.23 ± 0.83 mg L-1; p < 0.05)."

*The authors do not understand the above comment as not being new because these data presented here are new and unique. The reviewer has never referred to any specific citation of any peer reviewed study that quantifies pore water nitrate concentrations in alder hillslopes from permafrost landscapes. We agree that the fact that nitrate variability exists is not new but having quantitatively documented pore water nitrate variability (especially at fine spatial resolution) in these cryosystems is new. Yes, N has been measured in leaves and stream waters in Arctic Alder landscapes, but not directly in pore waters associated*

*with these plants. This work increases the referenceable literature in the community and allows for a better understanding of the degree of variability that can occur on short temporal and spatial scales in permafrost systems. The data presented in this study are new records of nitrate variability in a landscape/region (Seward Peninsula, AK) where Alder expansion is occurring, yet has been understudied. Our study will greatly aid in informing future studies in these systems because it demonstrates how variable nitrate concentrations are at the individual Alder patch scale as well as within and outside of Alder patches.*

I suggested that you express the nitrate as Nitrate-N = NO3-N (without a minus sign) to enhance the comparability to other N concentrations (e.g., NH4-N), but then you have to recalculate it:

For example: 4 mg L-1 NO3- = 0.90 mg L-1 NO3-N!

[The concentrations in the text and in the figures are NOW all wrong]
*Thank you for this constructive level of feedback. This comment is the result of a miscommunication on the last round of revisions. The authors originally included the data as $NO_3$-N but mis-referenced the concentrations within the text as $NO_3$-. The previous round of comments sparked the authors to realize this typo/mistake and so the terminology used within the text was updated to reflect the appropriate format, which is what the data was already presented in. We apologize for the confusion resulting from our original mistake and the need for additional details in communicating our previous round of revisions. Anonymous Referee #1 is completely correct that had we presented the numerical nitrate data as $NO_3$- originally, we would have needed to convert all our data. However, we originally included the numerical data in the correct form ($NO_3$-N) and mis-referenced it within our figures, tables, and text. This was an oversight that we are grateful that Anonymous Referee #1 identified that we display our data as 'nitrate-N' in the first round of revisions. The authors have verified the numerical data is indeed represented in the correct $NO_3$-N format by returning to and verifying our values within the raw and converted data. The authors apologize for the inclusion of the negative sign in the nitrate-N notation (formatting issue) and have gone through and removed the negative signs that were mistakenly included in our $NO_3$-N notation.*

Even correctly calculated, this difference can be expected as alder are N2-fixing plants (write N2-fixing not N-fixing).
*The authors have changed the notation to $N_2$-fixing plants as correctly identified by Anonymous Referee #1. While this finding could be 'expected', having additional values published that numerically document these spatial variabilities is still beneficial to the broader scientific community. A concept doesn't need to be new to be published and can still add value to the community through increasing data that exists on the topic.*

3) The design remains weak, when precipitation events seem to be the main driver for flushing events, but the precipitation has not been measured on site, there is no valid proof.
*While we do not have quantitative constraints on the precipitation event, the co-occurrence of precipitation on days where trends appeared could not be ignored. While the authors may have overstated the 'proof' of this in previous versions of the text, we are unable to separate the observed trends from our knowledge that precipitation events occurred on days where notable variability occurred. The authors have lessened the level of confidence in which we directly state the influence of precipitation on trends. We now state observed trends and indicate precipitation occurred on known days and that these variations are likely linked. See responses to line items left by Anonymous Referees #2-3 for more details of the changes to our text.*

4) Several references are still not cited correctly (Alnus viridis encroachment in the Alps occurs in the montane vegetation belt, not in the alpine, see Bühlmann et al. 2014, line 54) and more important: please make sure that you are not employing any plagiarism when using whole and/or "very similar" sentences from other articles (e.g., Salmon et al. 2019).

*The authors respectfully disagree with this point as the text in the instances Anonymous Reviewer #1 identifies have already been modified to more generalized landscapes that make these references appropriate within this context. For example, the Bühlmann et al 2014 reference in question follows text that states, '… have been investigated in alpine and upland systems…,' which goes beyond strictly alpine settings. However, we have further modified this text to state, "….have been investigated in alpine, subalpine, and upland systems…" and hope that is appeases Anonymous Reviewer #1 but we could also remove the Bühlmann et al 2014 reference completely if that would be preferred but we felt that it was a useful reference to direct readers to if they would like to explore studies that have already established relationships between alder and nitrogen chemistry.*

*Additionally, the Salmon et al 2019 study was co-located with this study and Salmon is a co-author on this manuscript. While we appreciate the level of concern for plagiarism, we firmly believe that the brief inclusion of modified text that best captures the vegetation descriptions within our study location, with the permission of the original author and a citation, is appropriate.*
* * *
**Anonymous Referee #2:**

Specific comments with line numbering

L 124: Is there a word missing between 'down' and 'gradient' ?
No word is missing. 'Down gradient' refers to water movement along the most likely flow pathway. In the instance of our sampling location, down gradient is usually synonymous with downslope.

L 126: Nice with the description of why you measured Mn and Fe :)
*Thank you kindly.*

L 145: 'hereafter' in stead of 'hereby'?
*Suggestion has been adopted.*

L 266: the (21-22) refers to the date, I assume – maybe change to (21-22nd) or (21-22.)
We have adopted the suggested (21-22$^{nd}$) reference style here and in other text where we reference dates (L 312).

L 283: Why is the labile N inaccessible according to Darouset-Nardi and Weintraub (2014)?
*The Darrouzet-Nardi and Weintraub 2014 article referenced here outlined that labile N could remain inaccessible if spatially isolated due to low water potential of an environment. We have added text (L 328-329) to communicate these details: "…Darrouzet-Nardi and Weintraub (2014) found evidence for spatial inaccessibility of labile N in Arctic ecosystems due to isolation in environments with low water potential, but…."*

L 291: If you want a more recent reference, Rasmussen et al. 2020 also saw a flush of organic C and N during and after a rainfall event.

*The authors thank Anonymous Referee #2 for this relevant reference suggestion and have incorporated it into the text.*

L 335-337: Agreed. And assessing the time around snowmelt and soil thaw and the transport potential related to redox environments present there is also necessary in the future.

*This is absolutely true. Perhaps there could be future collaborative potential in this area. We added text to section 4.5 Future Research (L 445-446) to denote that starting sampling/monitoring around snowmelt would be a beneficial future approach.*

Supplementary: A few of the references are underlined

*All references in the Supplement are now formatted appropriately.*
* * *
**Anonymous Referee #3:**

The aim of this study is to the N cycle and soil NO3- concentrations along a topographic gradient in a permafrost area with a vegetation cover of nitrogen fixing Alnus viridis spp fruticose. The main conclusion from the work emphasize the temporal variation in soil NO3- within and downslope from the alder shrub-land, where soil NO3- is supposedly being flushed downslope during precipitation events. This is an interesting and relevant topic of research to investigate N-cycle and transport-processes at landscape scales in permafrost regions, and the manuscript holds important data that I believe should be made available to the science community.

This manuscript has previously been revised in response to two extensive peer-reviews and I find that the authors have responded comprehensive and careful to the criticism raised. Meanwhile, I also find that the manuscript in it's current form can be further improved and streamlined to clarify and emphasize even stronger the outcome of this study. My main concern is specifically the speculation about precipitation driven downslope nitrate transport, combined with the isotopic observations that I recommend to moderate and soften. See comments below.

*The authors sincerely thank Anonymous Referee #3 for their thoughtful and constructive feedback on this manuscript. We appreciate the time and energy spent on these relevant, appropriate, and professional suggestions for improving the communication of our work and are grateful for the willingness of this referee to serve in this capacity. The type of feedback Anonymous Referee #3 provided exemplifies the benefits of a peer review process.*

Introduction

Line 31: A recent paper studies lateral N-transport in a permafrost landscape and demonstrates the function of lateral N-transport for plant uptake and growth. Rasmussen, L. H., et al. (2022). "Nitrogen transport in a tundra landscape: the effects of early and late growing season lateral N inputs on arctic soil and plant N pools and N2O fluxes." Biogeochemistry 157(1): 69-84.

*The authors thank Anonymous Referee #3 for this highly relevant reference and have incorporated it into the text accordingly.*

2.6 Statistical analyses

Line 194: The method for determining normal distribution needs to be mentioned.
*The authors have added text to provide these relevant details on L 213: "Data collected had normal distribution (identified through comparison of p-values to significance levels)..."*

Since data were all normally distributed, why was non-parametric test applied rather than more powerful parametric statistical tests?
*The authors chose to apply a non-parametric test after speaking with a statistical consultant who advised us that parametric approaches were unnecessary for our purposes. The non-parametric tests applied are valid for both normally and non-normally distributed data. Anonymous Referee #3 correctly identified that parametric tests on normally distributed data could provide additional insights to statistical interpretations. However, the statistical insights gleaned from the Mann-Whitney rank sum tests performed are appropriate for assessing our data for the purposed of this study.*

Line 205: Calcium, Sodium and Chloride statistical analysis is explained. Confusing as these ions have not been described in previous section on chemical analysis of soil water (only that cations and anions in general were analyzed)?
*The authors thank Anonymous Referee #3 for identifying this point of confusion. These statistical analyses were performed to compare common conservative chemical species ($Cl^-$ and $Na^+$) to a common non-conservative species (Ca) to gain insights to processes likely influencing the system. However, as noted, this text is distracting from the content of the main manuscript and these details were intended for inclusion in the Supplement. We have thus removed this text (and associated Brown 1998 reference) from the main article and included these details solely in the Supplement to support interpretations of processes influencing source of $NO_3^-$.*

Results

3.1 Soil depth and moisture, line 216: the equation percent dry/wet weight is not gravimetric soil moisture content but rather dry matter content. Please, specify or give correct equation.
*The authors thank Anonymous Referee #3 for bringing this oversimplification to our attention and have corrected the equation to: "((weight of wet soil – weight of dry soil) / weight of dry soil x 100)"*

3.2 Phase 1:… line 228: It's rather unfortunate that the local rain gauge malfunctioned at the time of the ongoing field work. Perhaps it could be relevant though to somehow provide indications for the amount of precipitation deposited during this event? The term …brief from the manuscript. Along this line, I also find the data on isotopic observations somewhat over-interpreted; e.g. no "real-time" data on nitrate in precipitation or isotopic values of nitrate and water in precipitation is presented to support the statement about precipitation isotopic imprints on soil water NO3. Also, with reference to Fig S2, is the predicted denitrification driven shift in isotopic values significant as this is based on very few data points? If not, I suggest to remove this line. (later in line 331 it's referred to as a trend in data).
*The authors agree and were quite disappointed by the timing of this malfunction that we only discovered after the fact. Unfortunately, the closest functioning quantitative rain gauge was over 60 km away from*

*our field location and due to geographical differences (coastal versus inland with topographic barriers in between) the data from this other location is not representative of the precipitation occurrence at our sampling location. There is no way to retroactively glean this information from our field site so we are left with our qualitative knowledge that rain occurred on certain days that we sampled. Because we do not have reliable records of the duration of rain or the volume of rain, we have removed the term 'brief' from the text since we cannot constrain the precipitation event with additional details.*

*The authors revisited the text with references to Fig. S2 and have removed the language that interprets isotopic results from the main text.*

*We also revisited text in section 4.2 and modified some of the language to lessen the confidence with which observed measurements were a direct result of precipitation events. Ex: we have modified 'associated with precipitation events" to "that co-occurred with precipitation events" and we have removed a sentence that overinterpreted the isotopic 'trend' from two sample points.*

*Additionally, we have added the following line of text to the supplement to clearly state that the co-occurrences in nitrate variability with precipitation events is not well constrained due to the lack of isotopic analyses from precipitation samples: "However, without the direct isotopic analyses of precipitation samples, these interpretations are based solely on the co-occurrence of precipitation events with observed chemical variability and are not quantitatively verified."*

4.3. Effects of redox...: Line 328: It's not clear to me how the lack of mobility of NO3 beyond the first 20-30 m downslope can be seen from Fig. 4 as this shows no downslope-scale (see comment above).
*The authors thank Anonymous Referee #3 for bringing the lack of spatial scale to our attention. We have added a line of text to the Figure 4 caption that adds these clarifying details: "Each sampling rhizon nest (depicted as histograms in these transects) is spaced ~10 m apart."*

4.4 Spatial and temporal... Line 344: As for my previous comments, I find the interpretations on soil NO3 in relation to precipitation basically unsupported as no precipitation data are presented. Moreover, in Fig. S1 it is shown that soil NO3 and soil moisture was not correlated, which somehow is conflicting the statement that "...the notable day-to-day changes in soil NO3...was driven primarily by the presence of rainfall."
*The authors have modified the language in this line from "correlated" to "co-occurred".*

*The authors have also modified the language, "the notable day-to-day changes in soil $NO_3$...was driven primarily by the presence of rainfall.", to "the notable day-to-day changes in soil $NO_3$...was likely influenced by the co-occurrence of rainfall as a mobilization mechanism"*

*With regard to Figure S1, we do not observe a correlation between soil moisture and nitrate concentrations in our study location. However, while the soil moisture samples are co-located with our soil pore water samples, they are not temporally correlated (collected during same sampling season but not same time as soil pore water samples). This is because sampling for gravimetric soil moisture content physically perturbs the system and could have influenced the pore soil water compositions if done at the same time. Thus, we collected soil moisture samples on our final days in the field to avoid influencing the chemistry of the water we collected and the resulting divergence between these parameters is likely a result of the temporal offset between sampling the different parameters. To clarify, the following text was added to Figure S1 caption:*

*"It is worth noting that soil moisture and NO₃-N samples were co-located but collected on different temporal scales to avoid perturbing the rhizon nests during our time-series sampling campaigns."*

*We have also added a line of clarifying text to section 2.4, "It is worth noting that while our soil moisture and soil pore water samples were co-located, they were offset temporally with our soil moisture sampling occurring at the end of our campaigns to avoid perturbing the rhizon nests while we were actively collecting soil pore water samples."*

*Thus, this lack of correlation between data collected at different times does not negate the variability in nitrate observed during or immediately following precipitation events at our study location.*

4.5 Future research. Line 357, see aforementioned paper by Rasmussen et al., 2022.

*The authors have incorporated this reference and amended the future work to reference the importance of snowmelt influence on NO₃ variability at the start of the grow season: "Future studies would benefit from the additional incorporation of continuous monitoring of NO₃⁻ throughout a growing season (starting around snowmelt; Rasmussen et al., 2022), …."*

Supplement

Isotopic insights: The inclusion of isotopic water and nitrate data is acknowledged as these can be important indicators for dominant processes in the present site. However, I do on the other hand also recommend data being interpreted and extrapolated with due attention to the fact that inputs and characteristics of e.g. NO3 in precipitation is not established in this study – see comment above.

*The authors appreciate this recommendation. While we were not able to measure the NO₃ in precipitation, we do have useful information regarding where our isotopic values plot relative to known compositions/ranges associated with likely sources of NO₃ within our environment. These interpretations are not the main focus of our body of work and this is thusly why this section has been moved to the Supplement rather than the main text. The authors have also added the following text to caution the reader of this possible issue: "These precipitation interpretations should be considered carefully as we were unable to directly measure the $\delta^{15}N$ and $\delta^{18}O$ of local precipitation due to logistical and sampling constraints. However, we apply logical inferences about likely influences of precipitation on observed chemical trends from our field location based on the known occurrence of rain events on specific sampling days and the corresponding shifts in site chemistry observed."*

Please, explain how 15N in TDN was measured.

*The following text was added before equations S2-S3 in the Supplement for clarity: "$\delta^{15}N_{TDN}$ was measured from the sample aliquots that underwent persulfate oxidation for TDN, which converts all ammonium, DON, and nitrite to nitrate. Thus, the $\delta^{15}N_{TDN}$ is from $\delta^{15}N$ analyses of the TDN sample aliquot."*

Coefficient of Variability: Table S6 (first line in section) should refer to Table S3.

*The authors have corrected the Table reference as suggested.*

Table S2. It's briefly mentioned (section 2.5) that DO, pH and conductivity didn't correlate with NO3-, but did the authors assess if there's any particular spatial or temporal pattern in variation of these parameters? Maybe worth to mention.

*The in situ parameters (DO, pH, and conductivity) collected from our field site were highly variable without any obvious spatial or temporal trends. The following line of text has been added to Table S2 caption and on L 204 within the main text: "no obvious spatial or temporal trends were observed within these parameters."*

---

## Author Response (AR3)

**Author Responses to Editor:**

*All author responses appear in grey italics below specific comments from the Editor. The authors thank the Editor for the time and energy they have spent facilitating reviews and providing constructive feedback on this body of work.*
* * *
**Editor Comments:**

Referee #3 Line 194: The method for determining normal distribution needs to be mentioned. Your response: The authors have added text to provide these relevant details on L 213: "Data collected had normal distribution (identified through comparison of p-values to significance levels)..." I believe referee asks for the specific test used for determining that data was normally distributed (e.g. a chi square test with $p < 0.05$, or similar information). Please provide some additional information in response to this comment.

*Upon further conversations with the first author, we identified that this line was unintentionally misleading as it was the isotopic data associated with this study that had normal distributions, but the concentration data associated with this study were non-normally distributed. With the transition of the isotopic work to the Supplement, all concentration data presented in the main text had non-normal distributions. Because Mann-Whitney rank sum tests were performed to assess significance, no statistical re-evaluation is needed as this test is appropriate for non-normal distributions. Corresponding edits have been made to section 2.6 to communicate that the concentration data had non-normal distributions and provide details of determining this. The first line of section 2.6 now reads: "Concentration data collected had non-normal distribution (identified visually via box plot asymmetry of concentration data) and no outliers were identified."*

Please also note the comment from review file validation: Please add the Last name "Wullschleger" to the co-author Stan D. in title page of *.pdf manuscript file.

*Our apologies for this unintentional deletion. The full names and affiliations of all authors are now verified.*

With this final revision I hope we can soon see the publication of this manuscript in TC, after a long review process.

*We thank the Editor for their facilitation and insights throughout this process and we also hope that this revision will result in the publication of this work.*